

# Heterogeneous Interactions between SO₂ and

# Organic Peroxides in Submicron Aerosol

Shunyao Wang[1], Tengyu Liu[2], Jinmyung Jang[1],

Jonathan P.D. Abbatt[2] and Arthur W.H. Chan[1*]

[1] Department of Chemical Engineering and Applied Chemistry, University of Toronto,

Toronto, Ontario, M5S 3E5, Canada

[2] Department of Chemistry, University of Toronto, Toronto, Ontario, M5S 3H6, Canada

*Correspondence to*: Arthur W.H. Chan (arthurwh.chan@utoronto.ca)





**Abstract**
Atmospheric models often underestimate particulate sulfate, a major component in ambient
aerosol, suggesting missing sulfate formation mechanisms in the models. Heterogeneous
reactions between $SO_2$ and aerosol play an important role in particulate sulfate formation and its
physicochemical evolution. Here we study the reactive uptake kinetics of $SO_2$ onto aerosol
containing organic peroxides. We present chamber studies of $SO_2$ reactive uptake performed
under different relative humidities (RH), particulate peroxide contents, peroxide types, and
aerosol acidities. Using different model organic peroxides mixed with ammonium sulfate
particles, $SO_2$ uptake coefficient ($\gamma_{SO2}$) was found to be exponentially dependent on RH. $\gamma_{SO2}$
increases from $10^{-3}$ at RH 25% to $10^{-2}$ at RH 71% as measured for a multifunctional organic
peroxide. Under similar conditions, the kinetics were found to be structurally dependent:
multifunctional organic peroxides have a higher $\gamma_{SO2}$ than those with only one peroxide group,
consistent with the reactivity trend observed previously in the aqueous phase. In addition, $\gamma_{SO2}$ is
linearly related to particle-phase peroxide content, which in turn depends on gas-particle
partitioning of organic peroxides. Aerosol acidity plays a complex role in determining $SO_2$
uptake rate, influenced by the effective Henry's Law constant of $SO_2$ and the condensed phase
kinetics of the peroxide-$SO_2$ reaction in the highly concentrated aerosol phase. These uptake
coefficients are consistently higher than those calculated from the reaction kinetics in the bulk
aqueous phase, and we show experimental evidence suggesting that other factors, such as
particle-phase ionic strength, can play an essential role in determining the uptake kinetics. $\gamma_{SO2}$
for different types of secondary organic aerosol (SOA) were measured to be on the order of $10^{-4}$.
Overall, this study provides quantitative evidence of the multiphase reactions between $SO_2$ and
organic peroxides, highlighting the important factors that govern the uptake kinetics.





**Introduction**

Sulfate and organic compounds are ubiquitous particulate components in both polluted and

pristine environments (Chen et al., 2009;Andreae et al., 2018;He et al., 2011;Sun et al.,

2013;Huang et al., 2014), with important implications for public health and global climate

(Hallquist et al., 2009). Particulate sulfate can form via S(IV) oxidation by OH radicals in the gas

phase and via oxidation in cloud water, fog droplets or the aerosol aqueous phase, including by

$H_2O_2$, $O_2$ (catalyzed by transition metals), $O_3$, $NO_2$ and small organic peroxides (methyl

hydroperoxide and peroxyacetic acid) (Seinfeld and Pandis, 2012). However, atmospheric

models tend to underestimate particulate sulfate production on both global (Tie et al., 2001;Yang

et al., 2017;Fairlie et al., 2010) and regional scales, especially during heavy haze episodes (Wang

et al., 2014;Zheng et al., 2015;Sha et al., 2019;Gao et al., 2016;Li et al., 2017;Huang et al.,

2019), suggesting that the overall kinetics may be underestimated and/or important mechanisms

may be missing in models.

To reconcile these differences, studies have investigated novel reaction mechanisms of sulfate

formation. Stabilized Criegee intermediates (sCIs) were found to oxidize $SO_2$ rapidly and

proposed to be an important source of ambient sulfate (Mauldin et al., 2012). When Liu et al.

(2019) applied this mechanism and kinetics to a source-oriented WRF-Chem model, the sCIs

pathway was found to only account for at most 9% of the total particulate sulfate. Reactive

nitrogen species (such as $NO_2$) have also been put forward to account for the missing sulfate at

relatively high aerosol pH (close to 7) (Wang et al., 2016;Cheng et al., 2016). However, such

high aerosol pH is not substantiated by thermodynamic models, which conclude that pH ranges

between 4 and 5 even in polluted regions (Song et al., 2018;Guo et al., 2017). A recent modeling

study incorporating this heterogeneous $NO_x$ mechanism still exhibited a discrepancy of 20%



between the predicted and observed sulfate, indicating the possibility of unknown mechanisms
(Huang et al., 2019). Other factors may play a role in enhancing the particle-phase sulfate
formation rates. Chen et al. (2019) investigated the synergistic effects of $NO_2$ and $NH_3$ on sulfate
formation, and found that the rate of this reaction can be enhanced by the high ionic strength in
the particle phase. This enhancement effect by solute strength on sulfate formation was also
investigated for the $H_2O_2$ pathway in aerosol liquid water. Liu et al. (2020) found ionic strength
and general acid-catalyzed mechanisms can cause the S(VI) formation rate to be nearly 50 times
faster in aerosol phase than in dilute solutions. On the other hand, during the severe haze
episodes in China (Li et al., 2020; Guo et al., 2017), transition metal ion (TMI) catalysis of $SO_2$
oxidation by $O_2$ can be significantly suppressed in the aerosol phase due to high ionic strength
(Liu et al., 2020;Cheng et al., 2016; Su et al., 2020).
In addition to high solute strength, submicron aerosol is also rich in organic compounds (Jimenez
et al., 2009;Hallquist et al., 2009). In recent years, many studies have investigated the potential
role of heterogeneous interactions between $SO_2$ and organic aerosol on particulate sulfate
formation. Song et al. (2019) found heterogeneous oxidation of hydroxymethanesulfonate
(HMS) by OH can trigger rapid sulfate formation. Wang et al. (2020) studied photosensitizers in
ambient particles and found this pathway could be essential under specific light conditions.
Recent studies found reactive intermediates from isoprene oxidation (Huang et al., 2019) and
benzoic acid (Huang et al., 2020), can yield a variety of organosulfur species upon catalysis by
TMI. Other studies have also investigated the interactions between secondary organic aerosol
(SOA) and $SO_2$. Field observations found that ambient sulfate abundance is highly correlated
with SOA formation (Yee et al., 2020;Xu et al., 2015). Liu et al. (2019) found that $SO_2$ enhances
SOA formation and average carbon oxidation state during methoxyphenol photooxidation. By



performing chamber experiments with limonene SOA formation in the presence of $SO_2$, Ye et al.
(2018) also observed significant $SO_2$ decay along with increased SOA yields and carbon
oxidation state, proposing that organic peroxides in SOA may be the key reactive intermediates
for $SO_2$ oxidation.
Organic peroxides are key intermediates for aerosol formation and ubiquitously exist in many
SOA systems (Hallquist et al., 2009;Bianchi et al., 2019). Numerous studies have reported
peroxide content of 20-60% for isoprene and monoterpene derived SOA (Surratt et al., 2006;Ng
et al., 2008;Ye et al., 2018;Epstein et al., 2014). A significant fraction of organic peroxide (30%-
50%) has also been found in naphthalene-derived SOA under low/high $NO_x$ conditions
(Kautzman et al., 2009). Using model simulations, Bonn et al. (2004) found organic
hydroperoxides can account for up to 60% of global SOA. The aqueous phase reaction kinetics
between organic peroxides and dissolved $SO_2$ have been explored in previous studies (Lind et al.,
1987;Gunz and Hoffmann, 1990;Wang et al., 2019;Dovrou et al., 2019;Yao et al., 2019). The
second order reaction rate constants for organic peroxides in SOA (Dovrou et al., 2019;Yao et
al., 2019) and S(IV) were measured to be on the order of $10^2$-$10^3$ $M^{-1}$ $s^{-1}$, which are within the
range of those measured for commercially available organic peroxides (Wang et al., 2019) and
small organic peroxides (Lind et al., 1987). Yao et al. (2019) quantified the reactive uptake
coefficient of $SO_2$ ($\gamma_{SO2}$) onto α-pinene SOA to be on the order of $10^{-4}$-$10^{-3}$, which is positively
dependent on RH and inferred particle-phase peroxide content. These reactions are also linked to
the formation of organosulfates (Wang et al., 2019). Both inorganic sulfate (85-90%) and
organosulfates (10-15%) were observed as products of $SO_2$ reactive uptake onto SOA (Yao et al.,

91    2019).



Given the potential significance of $SO_2$ reactive uptake in particulate sulfate formation, a more
in-depth study is needed to determine the important factors that govern the heterogeneous
kinetics of $SO_2$ onto organic peroxide containing aerosol. In this study, we measured $\gamma_{SO2}$ for two
categories of aerosol: 1. Model organic peroxides mixed with ammonium sulfate or malonic acid
and 2. SOA from a few representative biogenic and anthropogenic precursors. The impacts of
RH, peroxide type, peroxide content, and condensed phase pH on $SO_2$ reactive uptake were
evaluated systematically with the goal of better understanding atmospheric multiphase sulfate
formation.

**2. Methods**
The reactive uptake of $SO_2$ onto peroxide-containing particles was studied in a 1 m$^3$ Teflon
chamber under ambient temperature and pressure. In brief, generated particles and $SO_2$ were
introduced into the chamber separately. The consumption of $SO_2$, changes in particle size
distribution and chemical composition were monitored to estimate the reactive uptake
coefficients. Particles were also collected on filters for offline chemical characterization.

**2.1 Seed aerosol generation**
In this work, two types of aerosol were used to investigate the uptake of $SO_2$. The first is
ammonium sulfate or malonic acid mixed with model organic peroxides (Fig. S1). In this first set
of experiments, an aerosol atomizer (Model 3076, TSI Inc., USA) was used to generate aqueous
particles from dilute solution. Each solution consists of ammonium sulfate ($\geqslant$99%, Sigma-
Aldrich) or malonic acid (99%, Sigma-Aldrich) and a model organic peroxide in ultrapure water
(HPLC grade, Fisher Chemical). For the experiments investigating the relationship between $\gamma_{SO2}$





and peroxide type (Expt. 2-14), different commercially available organic peroxides were used,
including tert-butyl hydroperoxide (70 wt. % in water, Sigma-Aldrich), cumene hydroperoxide
(80 wt. % in water, Sigma-Aldrich), and 2-butanone peroxide (40% wt. % in water, Sigma-
Aldrich). The molar ratio of organic peroxide to ammonium sulfate in the atomizing solution was
2:1 with the aim of being atmospherically relevant (corresponding to maximum particulate
peroxide molar fraction of 66% and mass fraction of approximately 50-70% if all the organic
peroxides were assumed to remain in the particle phase). This ratio was used as a proxy for total
peroxide content in both gas and particle phase relative to that of ammonium sulfate upon
atomization. For the experiments studying the relationship between $\gamma_{SO_2}$ and particle-phase
peroxide content, the molar ratio of organic peroxide to ammonium sulfate (Expt. 10-12, 15-18)
in the solution was adjusted to be 0.02, 0.2, 1, 2, and 4, respectively. In experiments where
malonic acid was used (Expt. 19-22), molar ratios of 0.2, 1, 2, and 4 were adopted. For
measuring $\gamma_{SO_2}$ with different aerosol pH (Expt. 17, 23-25), different amounts of HCl (37%,
Sigma-Aldrich) were added into the solution prior to atomization. The atomized particles were
flowed into the chamber without drying, and therefore assumed to remain deliquesced under the
range of RH we studied. Expt. 2-14 also represent those where the relationship between $\gamma_{SO_2}$ and
RH conditions were studied.
In the second set of experiments, the uptake of $SO_2$ onto SOA was investigated (Fig. S2, Expt.
26-28). A custom-built 10 L quartz oxidation flow reactor was used to produce SOA (Ye et al.,
2016) from different hydrocarbon precursors. In this work, we studied SOA formed from toluene
photooxidation, limonene ozonolysis and α-pinene ozonolysis, 3 of the most commonly studied
SOA systems (Ng et al., 2007;Hildebrandt et al., 2009; Hartz et al., 2005;Varutbangkul et al.,
2006). Toluene (analytical standard, Sigma Aldrich) was injected continuously into zero air flow


by a syringe (1000 mL, Hamilton) installed on a syringe pump (KDS Legato100) to achieve an
initial concentration of 0.5 ppm. Limonene (Sigma-Aldrich, 97 %) and α-pinene (Sigma-Aldrich,
98 %) were pre-dissolved in cyclohexane (Sigma-Aldrich, 99.5 %) with a volumetric ratio of 1:
1500 and 1: 500 to ensure that OH formed from limonene or α-pinene ozonolysis is scavenged
by cyclohexane, estimated based on the rate constants (Atkinson and Arey, 2003). The initial
steady-state concentrations of limonene and α-pinene were controlled to be around 2 ppm and 1
ppm entering the flow tube. $O_3$, used as the oxidant (for limonene and α-pinene) or the OH
precursor (for toluene), was generated by passing 0.5 L min$^{-1}$ pure oxygen (99.6 %, Linde,
Mississauga, Canada) through an $O_3$ generator (no. 97006601, UVP, Cambridge, UK).
Humidified air was produced by bubbling zero air through a custom-made humidifier at a flow
rate of 1 L min$^{-1}$. The photolysis of $O_3$ produces O ($^1$D), which reacts with water vapour to
produce ·OH with illumination from the 254 nm UV lamps (UVP, Cambridge, UK) to initiate
the photooxidation of toluene. The average residence time inside the flow tube was controlled to
be around 5 minutes. A gas chromatography–flame ionization detector (GC-FID, model 8610C,
SRI Instruments Inc., LV, USA) equipped with a Tenax® TA trap was used to monitor the
concentration of hydrocarbon precursors at the inlet/outlet of the flow reactor. In all cases, the $O_3$
concentration was maintained to be at least 10 times higher than that of the hydrocarbon.
Temperature and relative humidity were monitored by an Omega HX94C RH/T transmitter.
Particle size distribution and volume concentration were monitored using a custom-built
scanning mobility particle sizer (SMPS), which is a combination of a differential mobility
analyzer column (DMA, model 3081, TSI, Shoreview, MN, USA) with flow controls and a
condensation particle counter (CPC, model 3772, TSI, Shoreview, MN, USA).



## 2.2 Quantification of $\gamma_{SO_2}$

Prior to each experiment, the chamber was flushed by purified air overnight with a flow rate of

25 L min$^{-1}$ until particle number concentration was less than 5 cm$^{-3}$ and SO$_2$ was less than 1 ppb.

To adjust RH, the chamber was humidified by passing purified air through a custom-built

humidifier filled with ultra-pure water. For experiments with atomized ammonium sulfate or

malonic acid, SO$_2$ was injected into the chamber prior to the introduction of particles. For

experiments studying $\gamma_{SO_2}$ onto SOA, aerosol generated from the flow tube was injected into the

Teflon chamber continuously after passing through an O$_3$ denuder (Ozone Solutions, Iowa, USA)

to achieve specific aerosol concentration inside the chamber prior to SO$_2$ addition. SO$_2$ mixing

ratio in the chamber during each experiment was continuously monitored using an SO$_2$ analyzer

(Model 43i, Thermo Scientific). The initial mixing ratio of SO$_2$ in each experiment was

controlled to be around 200 ppb. Aerosol size distribution was monitored by SMPS. The reactive

uptake coefficient of SO$_2$ was calculated by integrating the following equation:

$$-\frac{d[SO_2]}{dt} = \frac{1}{4}\gamma_{SO_2} A \bar{c} [SO_2] \tag{1}$$

Where [SO$_2$] is the SO$_2$ mixing ratio (ppb) monitored by the SO$_2$ analyzer; A is the average

surface area concentration ($\mu$m$^2$ cm$^{-3}$) derived from the particle size distribution measured by

SMPS; $\bar{c}$ represents the mean molecular velocity (cm s$^{-1}$) of SO$_2$. d[SO$_2$]/dt is solved over the

initial SO$_2$ decay, such that the peroxide concentration in the aerosol liquid phase is assumed to

be constant and pseudo-first order kinetics can be applied (Abbatt et al., 2012;Thornton et al.,

2003). A summary of all the measured $\gamma_{SO_2}$ can be found in Table S1. Typical evolution of

monitored species can be seen in Fig.1. Control experiments were performed in order to rule out

other potential factors (e.g. SO$_2$ loss in the in-line filter in front of the SO$_2$ analyzer, interferences

inside the SO$_2$ analyzer, chamber wall losses, SO$_2$ uptake onto wet ammonium sulfate, gas-phase



reaction of $SO_2$ with peroxide vapour) that may contribute to the $SO_2$ decay observed during the
$\gamma_{SO2}$ measurement inside the chamber (Fig. S3-S6). Measurement uncertainty of $\gamma_{SO2}$ in this study
was estimated from Expt. 10-12 to be 26%. Also, we observed there was $SO_2$ repartitioning from
the humid chamber wall in the presence of organic peroxide under high RH (Fig. S6b, RH 74%).
The observed $SO_2$ repartitioning rate was then applied to correct the $\gamma_{SO2}$ measured under high
RH conditions (above 70%, Expt.14), and this correction amounts to a 40% increase in
calculated $\gamma_{SO2}$.

**2.3 Offline peroxide quantification**
Aerosol was collected onto 47 mm PTFE (polytetrafluoroethylene) filters with 0.2 µm pore size
(Whatman®, GE Healthcare) from the chamber by a diaphragm pump (KNF Neuberger Inc.,
USA) for offline chemical analysis. The particulate peroxide content in these samples prior to
$SO_2$ uptake was quantified using the iodometric–spectrophotometric assay (Docherty et al.,
2005). $I_2$ produced from the reaction between $I^-$ and peroxides can further quickly combine with
the excess amount of $I^-$ to form $I_3^-$, which has brown color and absorbs UV-vis at 470nm. The
SOA extraction was then aliquoted into a 96-well UV plate (Greiner Bio-One, Kremsmünster,
AT) with 160 µL well$^{-1}$. 20 µL of formic acid ($\geq$ 95 %, Sigma-Aldrich) was added into each
well, following by 20 µL potassium iodide (BioUltra, $\geq$99.5%, Sigma-Aldrich) solution
(dissolved in DI water). The plate was then covered by an adhesive plate sealer (EdgeBio,
Gaithersburg, USA) immediately in order to avoid reagent evaporation and $O_2$ oxidation. After
incubation for an hour in the dark, the UV-vis absorption at 470nm was measured using a UV-
vis spectrophotometer (Spectramax 190, Molecular Devices Corporation, Sunnyvale, CA) and
then converted to peroxide concentration using the calibration curve made by tert-butyl



hydroperoxide (70 wt. % in $H_2O$, Sigma-Aldrich) with a series of concentrations (0-10mM). The
average molecular mass for aerosol was assumed based on the chemical composition in order to
calculate the molar fraction of total peroxides. More details about the iodometric-
spectrophotometric procedures were described in previous work (Wang et al., 2018).

**3 Results and discussion**
**3.1 $SO_2$ uptake and RH**
A positive relationship between $\gamma_{SO2}$ and RH (between 25 and 71%) was observed for all types of
organic peroxides studied (Fig. 2). The positive dependence of the reactive uptake coefficient of
water-soluble gaseous species on RH has also been observed in other studies (Thornton et al.,
2003;Griffiths et al., 2009;Zhao et al., 2017;Zhang et al., 2019). Recently, the uptake behavior of
$SO_2$ onto soot, mineral dust and SOA were also shown to positively depend on RH (Zhang et al.,
2019;Zhao et al., 2017;Yao et al., 2019).
It is also noteworthy that an exponential dependence of $SO_2$ reactive uptake coefficient on RH
was observed in our study. $\gamma_{SO2}$ increases with increased relative humidity, which could even be
more significant under high RH regime. This is consistent with previous laboratory studies that
measured the reactive uptake coefficient of $SO_2$ onto aerosol to be exponentially dependent on
RH (Zhang et al., 2019;Yao et al., 2019). Additionally, multiple field campaigns have observed
significant correlation between particulate sulfate formation and ambient RH (Song et al.,
2019;Sun et al., 2013;Huang et al., 2020). Sun et al. (2013) observed faster sulfate formation rate
under humid conditions, proposing a significant impact of aerosol liquid water on sulfate
production during wintertime in Beijing. Zheng et al. (2015) reported a notably higher SOR
(molar ratio of sulfate to the sum of sulfate and $SO_2$) during wet period (RH>50%), indicating



the importance of heterogeneous reactions to the secondary sulfur transformation with abundant
aerosol water content under humid conditions. In a recent study by Song et al. (2019), the rapid
sulfate formation rate observed under high RH conditions was found to be significantly higher
than atmospheric modeling results implemented with homogeneous $SO_2$ oxidation pathways,
which was later attributed to heterogeneous sulfate formation mechanisms. Multiple mechanisms
can potentially explain this observed $\gamma_{SO_2}$-RH dependence. An enhanced relative humidity would
result in a nonlinear increase of aerosol water content, which can lead to more $SO_2$ dissolved in
the aerosol aqueous phase (Seinfeld and Pandis, 2012). It should be noted that while the relative
humidity is varied systematically in these experiments, the relationship is more complex since
RH also affects other aerosol properties which can affect the uptake kinetics in turn. For
example, a higher aerosol liquid water content could dilute protons and thus lower the aerosol
acidity. In a study by Laskin et al. (2003), an enhanced uptake of $SO_2$ onto sea-salt particles was
observed with an increased aerosol alkalinity at high pH range.

**3.2 Dependence of $SO_2$ uptake on peroxide content and type**
As expected, the measured uptake rate of $SO_2$ is dependent on the particulate peroxide content in
the current study. Fig. 3 shows that $\gamma_{SO_2}$ is linearly proportional to the amount of particulate
peroxide for aerosol with similar volume-to-surface ratios and containing the same type of
organic peroxides. This positive relationship between $\gamma_{SO_2}$ and condensed phase peroxide content
has also been inferred from experiments of $SO_2$ uptake onto α-pinene SOA (Yao et al., 2019),
where the peroxide content in α-pinene SOA was varied indirectly by introducing NO and
adjusting the branching ratio of the peroxide-yielding $RO_2+HO_2/RO_2$ pathway.





In addition to the amount of peroxide injected, the particulate fraction of organic peroxide
available for heterogeneous reaction is also influenced by gas-particle partitioning. As indicated
in Fig. 2, the reactive uptake coefficients of different organic peroxides vary amongst each other
by about an order of magnitude in the range of RH studied, despite the same amounts of peroxide
relative to ammonium sulfate initially in the atomizing solution. Based on our previous work
(Wang et al., 2019), the aqueous-phase rate constants for these organic peroxides with dissolved
S(IV) only vary by a factor of 2-3 and therefore cannot fully explain the observed difference in
uptake rates. Since vapour pressure vary considerably among the different peroxides in the
present study, gas-particle partitioning is likely to influence the amount of peroxide in the
particle phase that react with dissolved $SO_2$. The relative particulate peroxide content on filters
of the three peroxides collected from chamber experiments under RH 50% without $SO_2$ uptake
were measured by the offline KI method (Fig. S7). Although the initial ratio of organic peroxide
to ammonium sulfate in the atomizing solution was nominally the same, we measured the highest
amount of particulate peroxide with 2-butanone peroxide (16.7%), followed by cumene
hydroperoxide (12.7%) and then tert-butyl hydroperoxide (3.8%) using the offline iodometric
method. This trend in particulate peroxide content is consistent with the vapour pressures
calculated using the SIMPOL group contribution method (Pankow et al., 2008), with 2-butanone
peroxide being the least volatile, and tert-butyl hydroperoxide being the most volatile. Also, the
order of particle-phase peroxide content is consistent with the order of $\gamma_{SO2}$ observed, as shown in
Fig. 2. A simple visualization of these relationships between different peroxide characteristics
(number of peroxide groups, vapour pressure and aqueous-phase rate constants) and measured
$\gamma_{SO2}$ (at RH = 50%) is illustrated in Fig. S7, which indicates higher $\gamma_{SO2}$ can be expected for
multifunctional organic peroxides with lower vapour pressures and higher aqueous phase



reactivities. It should be noted that the order of magnitude difference in experimentally measured
$\gamma_{SO2}$ among various organic peroxides (Fig.2) is still not fully explained when both volatility and
reaction kinetics are taken into account (Fig.S7), suggesting that the reactive uptake may be
influenced by other factors. In summary, for our current experiments where we nominally
maintained total injected amount of organic peroxide constant, measured $\gamma_{SO2}$ depends both on
reactivity and gas-particle partitioning of the organic peroxides.

**3.3 SO$_2$ uptake and aqueous phase kinetics**
Since the aqueous phase reaction rate constants between S(IV) and these model organic
peroxides have been measured previously (Wang et al., 2019), we can test our understanding of
the measured $\gamma_{SO2}$ using a simple model. By assuming the amount of SO$_2$ dissolved in the aerosol
is in equilibrium with the gas phase, the overall $\gamma_{SO2}$ can be expressed using the simplified
resistor model (Hanson et al., 1994):

$$\frac{1}{\gamma} = \frac{1}{\alpha} + \frac{\bar{c}}{4HRT\sqrt{k^I D_l}} \frac{1}{\left[\coth(q) - \frac{1}{q}\right]}$$    (2)
where $\alpha$ is the mass accommodation coefficient, $\bar{c}$ is the mean molecular speed of SO$_2$ (cm s$^{-1}$),
H is the effective Henry's law constant that includes both the dissolution of SO$_2$ and the
dissociation of H$_2$SO$_3$ (M atm$^{-1}$), R is the ideal gas constant (atm L mol$^{-1}$ K$^{-1}$), T is the
temperature (K), and the parameter q is used to describe the competition between the reaction
and diffusion of the dissolved gaseous species within a particle, which is further calculated as:
$$q = r\sqrt{\frac{k^I}{D_l}}$$    (3)
where r is the radius (cm) of a given particle, D$_l$ is the aqueous-phase diffusion coefficient (cm$^2$
s$^{-1}$), k$^I$ is the first order rate constant (s$^{-1}$) for the reaction. For experiments in the current study,



the calculated q values were consistently found to be far less than 1, which indicates a volume-
limited reaction regime. Combining with the assumption of a relatively fast mass
accommodation process compared with the bulk phase reaction, equation (2) can be further
simplified as to describes reactive uptake in the volume-limited regime:
$$\gamma = \frac{4HRT[peroxide]k^{II}}{\bar{c}}\frac{V}{S} \qquad (4)$$
Here, we assume all the peroxides remain in the condensed phase upon atomizing and reaction
inside the chamber for the upper-bound prediction of $\gamma_{SO2}$. [peroxide] represents the particle phase
concentration of total organic peroxide (M) based on the initial ratio between organic peroxide and
ammonium sulfate in the atomizing solution, and the aerosol water content output by E-AIM III
(Clegg et al., 1998), $k^{II}$ is the second order reaction rate constant ($M^{-1}$ $s^{-1}$), which we have
measured in the bulk phase at dilute concentrations previously (Wang et al., 2019), V/S is the ratio
between particle volume concentration ($\mu m^3$ $cm^{-3}$) and particle surface area concentration ($\mu m^2$
$cm^{-3}$) derived from SMPS measurements. As a result, the observed reactive uptake coefficient of
$SO_2$ can be compared to that predicted from the bulk phase reaction rate constant, and the results
are shown in Fig. 4 and Fig. S8. Overall, we noticed that this model captures the dependence of
$\gamma_{SO2}$ on peroxide content, but the modeled results were found to be generally 15-50 times lower
than the experimentally measured values (Fig. S8). The current $\gamma_{SO2}$ predictions are likely upper-
bound estimates since all the peroxides were assumed to stay in the condensed phase without
partitioning. As a result, this observed 15-50 times of discrepancy could even be larger if the
particulate peroxide content during the chamber experiments were lower due to partitioning.
It should be noted that the calculated $\gamma_{SO2}$ was based on reaction kinetics measured in dilute
solutions while the experimental $\gamma_{SO2}$ were measured directly from suspended particles. This large
difference in kinetics between those in aerosol and in dilute bulk solution suggests that this



multiphase interaction is strongly favored in the highly concentrated aerosol environment. One of
the potential explanations for this discrepancy could be liquid-liquid phase separation (LLPS) in
aerosol between organic peroxide and ammonium sulfate (Ciobanu et al., 2009;O'Brien et al.,
2015) such that $SO_2$ can directly interact with the acidic organic phase, where the concentration of
peroxides can be higher and the kinetics can be different from what we have measured in dilute
solution (Wang et al., 2019). However, LLPS is generally governed by the chemical composition
of the hydrophobic phase (Freedman, 2017). A higher level of oxygenation in organic aerosol is
related with higher hydrophilicity, which would favor a homogeneous particle instead of phase
separation. Previous studies showed that LLPS did not occur for organic coating with O:C above
0.8 (You et al., 2013;You et al., 2014). The LLPS phenomenon in simple organic–inorganic
mixtures can also be affected by the functional groups. The maximum O:C for LLPS could be 0.71
for organics with multiple carboxylic and hydroxyl groups but low aromatic content (Song et al.,
2012) while the 2-butanone peroxide we used for both $\gamma_{SO2}$ measurement and prediction in the
present study has multiple peroxide groups with an O:C value of 0.75. Particle size could also have
impacts on phase separation (Cheng et al., 2015). Particle diameters in the current study are mainly
under 200 nm while a previous study showed particles smaller than this size are less likely to
experience LLPS (Veghte et al., 2013). We therefore believe that LLPS is not likely to be
responsible for the enhanced uptake rate observed under these experimental conditions.
Another explanation is the high solute strength in the concentrated aerosol phase. As indicated in
Fig. 4 and Fig. S8, the difference between the measured and predicted $\gamma_{SO2}$ is larger for ammonium
sulfate aerosol than for malonic acid. Meanwhile, the calculated ionic strength in aerosol liquid
phase under RH 50% for ammonium sulfate (40 mol kg$^{-1}$) is significantly larger than that of
malonic acid (0.45 mol kg$^{-1}$). It has been previously reported that the reaction rate between sulfite





and hydrogen peroxide in aqueous phase increases with ionic strength (Maaß et al., 1999). Based
on the reaction mechanisms proposed for dissolved $SO_2$ and hydrogen peroxide (Halperin and
Taube, 1952), we speculate the reaction between aqueous phase S(IV) and organic peroxides to
follow a similar mechanism:

$$\underset{HO}{\overset{O}{\underset{\|}{S}}}\underset{O^-}{} + ROOH/ROOR \quad \overset{K}{\rightleftharpoons} \quad \beta \overset{O^-}{\underset{O}{>}} S-O-O-R + (1-\beta)\,SO_4^{2-} + H_2O/ROH \qquad (5)$$

$$O^--\overset{\|}{\underset{\|}{S}}-O-O-R \; + H^+ \quad \overset{K_a^{-1}}{\rightleftharpoons} \quad HO-\overset{O}{\underset{\|}{S}}-O-R \qquad (6)$$

$$HO-\overset{O}{\underset{\|}{S}}-O-R \quad \overset{k}{\longrightarrow} \quad O^--\overset{O}{\underset{\|}{S}}-O-R \; + H^+ \qquad (7)$$

where the overall rate constant is equal to $k\frac{K}{K_a}$, assuming fast equilibrium steps for reactions 5 and
6. Dissociated solutes are surrounded by an extended solvation shell which could affect the
reaction rates (Herrmann, 2003). Fewer available free water molecules would therefore shift the
equilibrium to the right in equation (5). Additionally, higher ionic strength also corresponds to an
increased concentration of electrolytes in the aqueous phase, which could hinder the dissociation
of the peroxymonosulfurous acid and shift the equilibrium in equation (6) to the right. In recent
work by Liu et al. (2020), the rate of S(IV) oxidation by $H_2O_2$ can be enhanced by up to a factor
of 50 in aerosol aqueous phase compared to that of dilute solution. The highest ionic strength at
which such enhancement was measured for the $H_2O_2$ oxidation pathway was 15 mol kg$^{-1}$ (Liu et
al., 2020).
Whereas the above analysis is based on the assumption that all the chemistry occurs in the bulk
component of the particle, it is also possible that some component of the reaction occurs at the gas-
particle interface and the overall kinetics can be affected by interfacial characteristics. For example,
an enhanced ionic strength in the aerosol phase can also impact the interfacial reaction mechanisms.
Previous study has shown evidence that interfacial chemistry is important for $SO_2$ oxidation in the





aerosol phase (Laskin et al., 2003). With higher ionic strength, anions partitioning to the air-liquid
interface can promote the overall reaction kinetics via proton transfer and thus accelerate the
interfacial chemistry (Knipping et al., 2000;Mishra et al., 2012;Mekic et al., 2018;Mekic et al.,
2020; Wei et al., 2018; Ruiz-Lopez et al., 2020). However, it should be noted that there is no
evidence from the current study showing direct relationship between the interfacial properties and
$\gamma_{SO2}$, and future studies are warranted.
Therefore, while more studies are needed to clearly delineate the roles of ionic strength, interfacial
activity, bulk reactivity, and particle phase state quantitatively, the enhancement of $SO_2$ oxidation
kinetics by highly concentrated aerosol particles compared to dilute aqueous solutions are
concluded to be large (factor of 15-50) for the experimental conditions in the current study.

**3.4 $SO_2$ uptake and aerosol pH**
As indicated by the proposed reaction mechanisms (Eqn. 5-7), protons are important reaction
intermediates for this $SO_2$ oxidation pathway. Previously, the aqueous phase reaction rate
constants between organic peroxides and dissolved $SO_2$ were measured to be pH dependent
(Wang et al., 2019). Moreover, the dissolution equilibrium of $SO_2$ into aqueous phase is also pH
sensitive (Seinfeld and Pandis, 2012). Besides, many studies have shown that the uptake kinetics
for gaseous species can be affected by the condensed phase pH (Shi et al., 1999;Gaston et al.,
2014;Drozd et al., 2013;Jang and Kamens, 2001;Liu et al., 2015). Reactive uptake of ammonia
was observed to depend on condensed phase acidity (Shi et al., 1999). Heterogeneous
condensation of isoprene-derived epoxydiol onto seed aerosol was found to increase with proton
concentration (Gaston et al., 2014). In the current study, the potential impact from particle phase
pH on $\gamma_{SO2}$ was explored by adding HCl into the atomizing solution. To estimate the particle





phase pH, two different methods associated with two different assumptions were used. In the
first scenario, the aerosol pH in each experiment was estimated using the E-AIM III model
(Clegg et al., 1998) based on the initial molar ratios of inorganic species ($H^+$, $NH_4^+$, $SO_4^{2-}$, $Cl^-$) in
the atomizing solution and measured RH (around 50%). In the second scenario, the additional
sulfate formed from reactive uptake of $SO_2$ was taken into consideration. The partitioning of HCl
was allowed in the model simulation for both scenarios. The formation of sulfate would enhance
the proton concentration in the aerosol liquid phase thus lower the aerosol pH. The average pH
during the $SO_2$ uptake process is likely in between these two extremes.
Fig. 5 shows the measured reactive uptake coefficients of $SO_2$ as a function of the calculated pH.
The reactive uptake coefficient was found to increase with increasing proton concentrations
(decreasing pH), which is consistent with acid-catalyzed reactions between peroxides and
dissolved $SO_2$ as measured in the bulk phase (Lind et al., 1987; Wang et al., 2019). $\gamma_{SO2}$ was also
predicted for the same range of pH based on Eqn. 4 and the pH-dependent bulk-phase reaction
rate constants measured previously (Wang et al., 2019). Indicated by Fig. 5, the measured $\gamma_{SO2}$
again exceeds the predicted $\gamma_{SO2}$ by about a factor of 50, which is consistent with what we
reported earlier and is likely due to the effects of aerosol ionic strength.
Unlike the observed $\gamma_{SO2}$, however, the predicted $\gamma_{SO2}$ does not exhibit a monotonic trend. $\gamma_{SO2}$ is
expected to decrease with decreasing pH at high pH (>2) as the effective Henry's law constant of
$SO_2$ decreases with higher acidity (Seinfeld and Pandis, 2012). $\gamma_{SO2}$ is not expected to increase
with decreasing pH until pH is below 2 where the acidity enhancement in reaction rate constant
exceeds the decrease in $SO_2$ solubility. As illustrated earlier, extrapolating dilute aqueous-phase
kinetics to the highly concentrated aerosol requires considering effects from high solute strength.
Solute strength may change the pH dependence of $\gamma_{SO2}$ in two ways. First, the solubility of $SO_2$





may decrease and become less dependent on pH as ionic strength increases (Rodríguez-Sevilla et
al., 2002). A former study (Leng et al., 2015) has shown that the effective Henry's law of
triethylamine decreases with increased ionic strength. Another potential explanation is that the
aqueous phase reaction rate constant can be more pH-dependent at high ionic strengths than what
we measured previously in dilute solutions. In either case, the inflection of the predicted $\gamma_{SO_2}$
would change and $\gamma_{SO_2}$ could become more negatively dependent on pH ($d[\gamma_{SO_2}]/d[pH]$ becomes
less positive in the high pH range and/or more negative in the low pH range), which would
match more closely with the observed dependence. It should also be noted that there are
substantial uncertainties in estimating pH values since the reactive uptake is a dynamic process
and will influence aerosol pH in turn upon sulfate formation. In summary, while the magnitude
of predicted $\gamma_{SO_2}$ is consistent with our expected values (after accounting for the enhancement by
high aerosol solute strength), we cannot fully explain the dependence of $\gamma_{SO_2}$ on aerosol pH at the
current stage. Future studies should investigate how the effective Henry's law of $SO_2$ and pH
dependence of reaction rate constants vary in aerosol liquid phase with high solute strength in
order to have a more comprehensive understanding of the relationship between $\gamma_{SO_2}$ and aerosol
pH.
**3.4 $SO_2$ uptake onto SOA**
$\gamma_{SO_2}$ was measured for a few model SOA systems, as organic peroxides are abundant in SOA
(Surratt et al., 2006;Kautzman et al., 2009;Krapf et al., 2016;Bonn et al., 2004). Here we studied
SOA formed from monoterpene ozonolysis and toluene photooxidation. It should be noted that
for the $\gamma_{SO_2}$ measurements of toluene SOA, a strong hydrocarbon interference was observed with
the $SO_2$ analyzer, likely stemming from the high concentrations of gas-phase aromatic
compounds. A rough estimate of the uptake rate for toluene SOA from aerosol mass



spectrometer sulfate measurements is provided in the SI (Section 1). The reactive uptake
coefficient of $SO_2$ onto Saharan mineral dust was reported on the order of $10^{-5}$ (Adams et al.,
2005). $\gamma_{SO2}$ onto dust with the coexistence of $NO_2$ and $NH_3$ under various RH conditions were
measured to be $10^{-7}$ to $10^{-5}$ (Zhang et al., 2019). For a variety of metal oxides, $SO_2$ reactive
uptake coefficients were quantified to be between $10^{-6}$ and $10^{-4}$ (Usher et al., 2002; Fu et al.,
2007; Shang et al., 2010). More recently, $\gamma_{SO2}$ studied for heterogeneous sulfate formation by
photolysis of particulate nitrate were reported in the range of $10^{-6}$ to $10^{-5}$ (Gen et al., 2019). As
shown in Fig. 6, $\gamma_{SO2}$ for all SOA systems were measured to be on the order of $10^{-4}$. Similar $\gamma_{SO2}$
values on the order of $10^{-4}$ were measured for α-pinene SOA by Yao et al. (2019), and $10^{-5}$ for
limonene SOA estimated from the chamber study by Ye et al. (2018). The reaction products
from this SOA and $SO_2$ interaction will be reported in a separate study.

**4. Atmospheric Implications**
Oxidation of atmospheric hydrocarbons produces reactive intermediates that can potentially
interact with $SO_2$ and form particulate sulfate, contributing to PM formation and growth (Berndt
et al., 2015;Mauldin et al., 2012;Yao et al., 2019). Organic peroxides generated from both
biogenic and anthropogenic hydrocarbon emissions are abundant in submicron aerosol. Given
that they are highly reactive with relatively short lifetimes (Bonn et al., 2004;Krapf et al.,
2016;Qiu et al., 2020), these species could serve as important condensed phase oxidants for gas
phase $SO_2$. Combining laboratory measurements and model predictions, the current study
investigated heterogeneous reactions between $SO_2$ and particulate organic peroxide. The
measured $\gamma_{SO2}$ for organic peroxide containing aerosol ranges from $10^{-5}$ to $10^{-2}$ in this study.
Based on the modeling work by Wang et al. (2014), adding an $SO_2$ uptake pathway to GEOS-



Chem with a reactive uptake coefficient of $10^{-4}$ could improve the surface sulfate prediction by
more than 50% during a severe haze episode over North China (RH 50%), suggesting the
potential importance of this multiphase reaction pathway, especially when SOA is the dominant
component in particulate matter.
The dependence of the heterogeneous kinetics on RH, aerosol pH, peroxide type, and peroxide
content were also evaluated. The experimentally measured $\gamma_{SO2}$ was found to be consistently
higher than that predicted from reaction kinetics with organic peroxides in the dilute aqueous
phase. This discrepancy can be potentially explained by the effects of high ionic strength
presented in the aerosol, suggesting that the impact from highly concentrated solutes needs to be
taken into consideration when applying aqueous phase kinetics to aerosol multiphase chemistry,
especially for particles containing strong electrolytes. We also observed that the kinetics of this
multiphase reaction exhibit a weak dependence on pH. Increasing the condensed-phase acidity
enhances the heterogeneous rate constant at low pH range, and while this pH dependence is
consistent with that of the aqueous phase reaction rate constant measured previously, it is not
consistent with the decrease of effective Henry's law constant of $SO_2$ along with enhanced
acidity. Currently, we are not able to fully explain the pH dependence, likely due to the
uncertainties from high ionic strength, and further studies are warranted. Particle phase peroxide
content was observed to be linearly correlated with $\gamma_{SO2}$. Moreover, $\gamma_{SO2}$ measured for 2-butanone
peroxide was found to be orders of magnitude higher than that of cumene hydroperoxide and
tert-butyl hydroperoxide. The difference in $\gamma_{SO2}$ among various types of organic peroxides can be
partially explained by their condensed-phase reactivity and gas-particle partitioning.
In general, we found the observed $\gamma_{SO2}$ in this study can be summarized using the following
semiempirical multilinear relationship:





$$\log \gamma = -1.7 + 0.0024 \times k^{II} + 0.46 \times PAS + 0.024 \times RH - 1.9 \times Vp \qquad (8)$$
where $\gamma$ is the reactive uptake coefficient, $k^{II}$ is the aqueous phase S(IV) oxidation rate constant
(M$^{-1}$ s$^{-1}$) , $PAS$ is the molar ratio between particulate peroxide and ammonium sulfate in the
atomizing solution, which is a proxy for the amount of peroxide in both gas and particle phases
applied in the current study, $RH$ is the relative humidity (%),$Vp$ is the vapour pressure (kPa) of
the peroxide. Fig. 7 illustrates the degree to which this semi-empirical expression describes the
experimental data for ammonium sulfate aerosol mixed with the three types of organic peroxides.
Residual evaluations of this multilinear regression can be found Fig. S9. We caution that this
equation is not directly applicable to atmospheric models in its current form, especially since the
particle phase peroxide content (PAS) value we applied as input is a calculated value, rather than
a measurement. However, it illustrates the internal consistency of our experimental results across
a range of RH, peroxide content, and aqueous phase reactivities, which are the key variables for
uptake rates. Better understanding of ionic strengths and pH in aerosol, either through modeling
or direct measurements of these variables, is needed to establish the coefficient dependence.
Future studies should be focused on exploring $\gamma_{SO2}$ and the reaction products for various types of
SOA as well as ambient particles under atmospherically relevant conditions, evaluating the
underlying impacts from photochemical condition, chemical composition, particle morphology,
ionic strength and interfacial properties on this multiphase physicochemical process. Overall,
$\gamma_{SO2}$ presented in our study and its relationship with ambient RH, aerosol pH, ionic strength,
particulate peroxide content and type could provide a framework for the implementation of this
heterogeneous mechanism in atmospheric models to have a better understanding of ambient
sulfate formation and particle growth.



*Author contributions*
A.W.H. C. and S.W. designed the study. S.W., T. L., and J. J. performed the experiments. S.W.,
A.W.H. C., T. L., and J. J. analyzed data. S.W. and A.W.H. C. wrote the manuscript with the
input from all co-authors.

*Data availability*
All data presented in this study are available in the supplemental material and have been
deposited in figshare.

*Associated content*
Supporting Information.

*Competing interests*
The authors declare no competing financial interest.

*Acknowledgements*
This work was supported by Natural Sciences and Engineering Research Council Discovery Grant.
The authors would like to thank Dr. Greg Evans, Dr. Yue Zhao and Dr. Christopher Lim for helpful
comments and discussions. Special thanks to SOCAAR for providing the $SO_2$ analyzer.









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




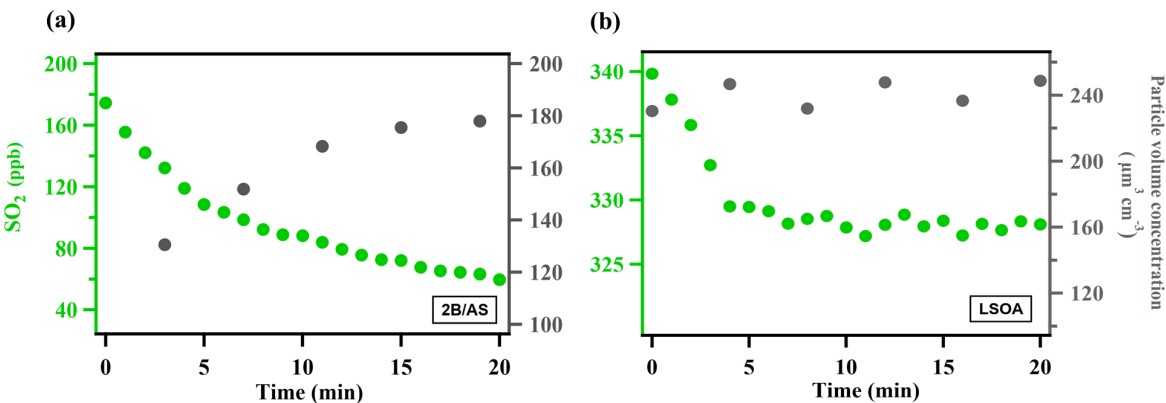


**Figure 1.** Typical evolution of the species monitored during $\gamma_{SO2}$ measurement for (a)

ammonium sulfate mixed with 2-butanone organic peroxide (2B/AS, Expt. 16) and (b) limonene

SOA (LSOA, Expt. 27). Particle volume concentrations measured by SMPS have been corrected

for wall loss assuming a pseudo first-order loss rate (Ye et al., 2016). $\gamma_{SO2}$ was calculated for the

initial portion of the decay (first 7 minutes).









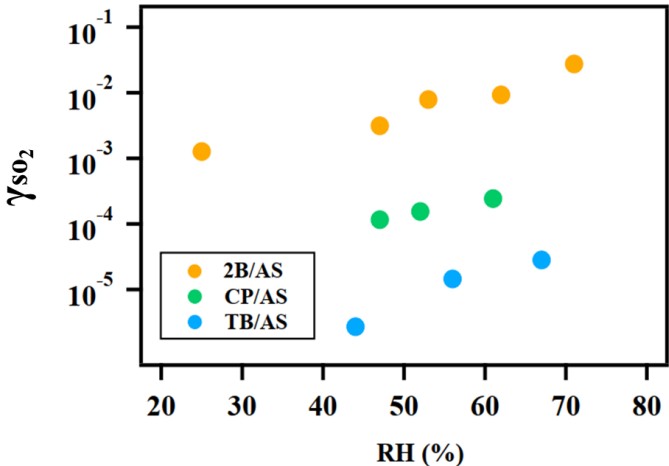


**Figure 2.** Exponential relationship between $\gamma_{SO2}$ and RH for ammonium sulfate aerosol

containing 2-butanone peroxide (2B), cumene hydroperoxide (CP), tert-butyl hydroperoxide

(TB).













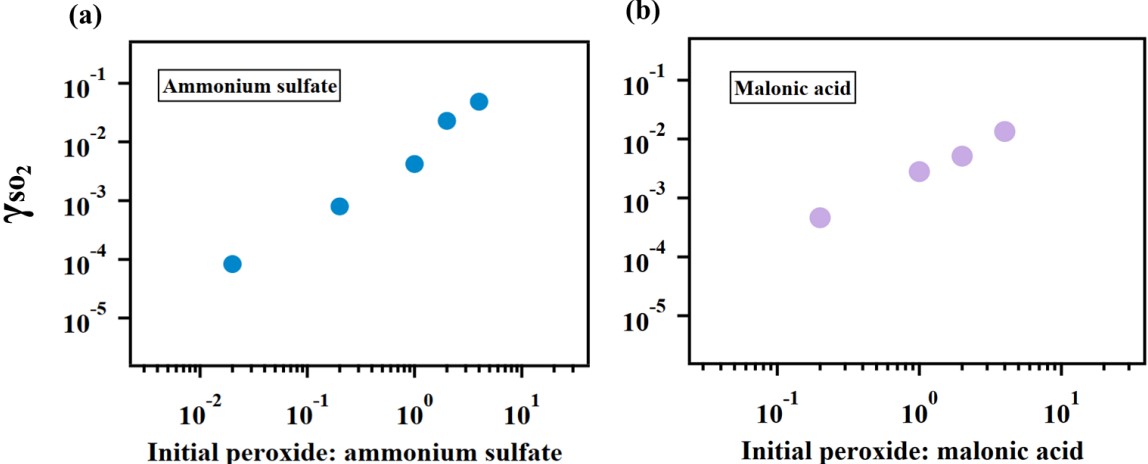


**Figure 3.** Relationship between $\gamma_{SO2}$ and particulate peroxide content. $\gamma_{SO2}$ for ammonium sulfate
(a) and malonic acid aerosol (b) containing different amount of 2-butanone peroxide are shown
here. The observed dependence of $\gamma_{SO2}$ on the amount of peroxide injected are linear since the
slopes of the relationship are both nearly 1 in (a) and (b).







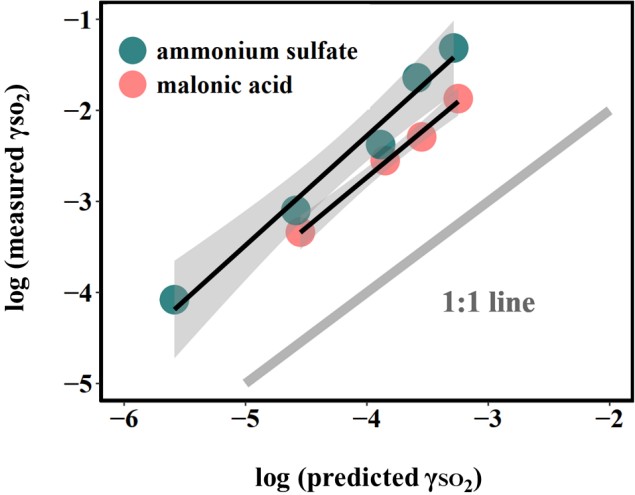


**Figure 4.** Relationship between measured $\gamma_{SO2}$ and $\gamma_{SO2}$ predicted by Eqn. 4. The large deviation

from the 1:1 line, which represents the difference between the measured uptake coefficient and

predicted values based on kinetics in the dilute aqueous phase, indicates that aerosol reactive

uptake is significantly faster than reactions in dilute aqueous phase. This enhancement is likely

driven in part by high ionic strengths, as the difference between measured $\gamma_{SO2}$ and predicted $\gamma_{SO2}$

are consistently higher for organic peroxide containing ammonium sulfate (high ionic strength)

than for that mixed with malonic acid (lower ionic strength).








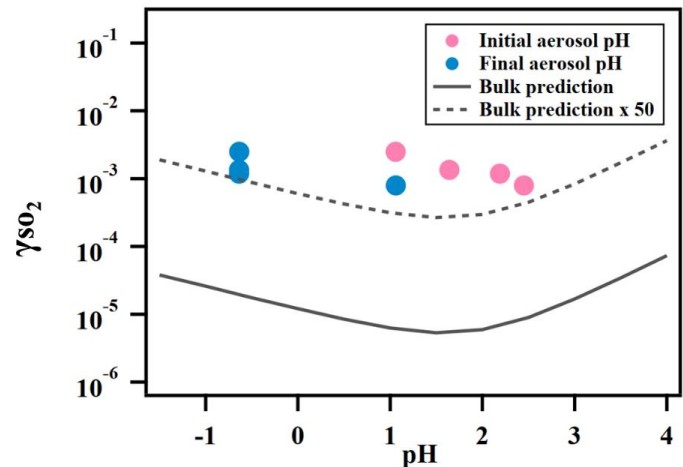


**Figure 5.** Relationship between $\gamma_{SO_2}$ and aerosol phase pH for ammonium sulfate aerosol

containing 2-butanone peroxide.













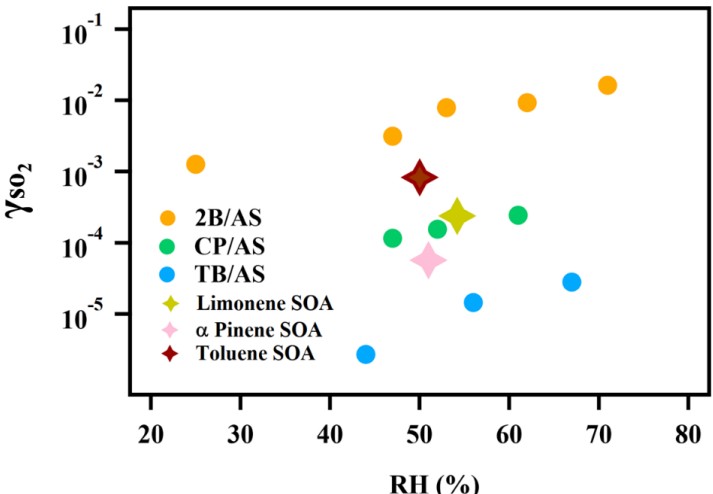

**Figure 6.** $\gamma_{SO2}$ measured for different types of organic aerosol. The reactive uptake coefficient of
SO$_2$ onto SOA are on the order of 10$^{-4}$.













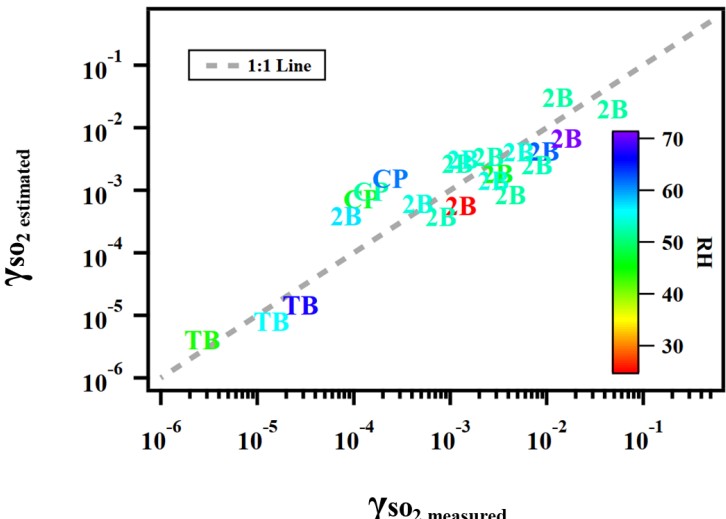


**Figure 7.** Predicted $\gamma_{SO2}$ using Equation (8) versus measured $\gamma_{SO2}$ for ammonium sulfate or

malonic acid aerosol containing 2-butanone peroxide (2B), cumene hydroperoxide (CP), tert-

butyl hydroperoxide (TB) under different experimental conditions.






