# Peer review of "Heterogeneous Interactions between SO2 and Organic Peroxides in Submicron Aerosol"

_Atmospheric Chemistry and Physics, 2020_

## Referee Comment (RC1) · Anonymous Referee #1 · 28 Oct 2020

The authors studied the uptake coefficients of sulfur dioxide on particles containing three model organic peroxides (tert-butyl hydroperoxide, cumene hydroperoxide, and 2-butanone peroxide) as a function of (a) relative humidity (as proxy for particle liquid water), (b) particle acidity, and (4) composition of the particles (e.g., with malonic acid, or ammonium sulfate, or various model SOA material generated under dry conditions). The SO2 was measured by a commercial analyzer and the particles were measured by SMPS. The pH was modeled by E-AIM. The methods are sound, and the paper is well written, and the discussion is fairly thorough. Moreover, the results are likely important for global modeling to better understand the atmospheric sulfur cycle. I request minor revisions based on the specific comments below.

Specific comments (line number precedes comment)

[Figure]

12 The terminology is a bit confusing. As I understand it "multifunctional" means multiple different functional groups (e.g., an alcohol and a hydroperoxide on the same compound) not multiple peroxide groups. Perhaps multiple peroxide groups on a compound would be better described as poly-peroxide (similarly to polyol) or just multiple peroxide. Please clarify this throughout the text. Also please be specific throughout the text whether you are referring to hydroperoxide moieties or all peroxides.

12 As the authors only studied three peroxides, and they are not analogues in the way that would make the hydroperoxide moiety dependence clear, I would suggest against generalizing with this statement. At least the authors should add "in this study" to the statement to avoid overly broad generalizations, or revise in another way.

39 Mauldin et al 2012 did not positively identify stabilized Criegees, they suggested that it was a "Compound X" or "Unexplored oxidant X" that they believe to be SCIs. However, kinetic competition studies between SCI and water vapor vs SO2 found that SCI + SO2 is not competitive in the atmosphere for the dominant SCI CH2OO (Newland et al, ACP 2015, Nguyen et al, PCCP 2016). It is not clear which rates are used in Liu but that study seems to back up the previous lab work, as Nguyen et al estimated that CH2OO alone would be responsible for <6% SO2 oxidation at a Southeast US site. I suggest to change this sentence to "sCIs were hypothesized to oxidize..." and please acknowledge the works before Liu 2019 that have already shown this pathway to be non-competitive at realistic RH using lab studies. https://core.ac.uk/download/pdf/267289280.pdf https://pubs.rsc.org/en/content/articlepdf/2016/cp/c6cp00053c

115 (and elsewhere) The authors should insert the SI table number explicitly after the Experiment numbers so the reader can know where to look up the experiments.

127 Please state the "different amounts" of HCl added for each experiment and the estimated particle pH that the different amounts of HCl correspond to. The authors say later that they estimate particle pH using E-AIM but this is worth mentioning in methods

briefly first.

184 Where the losses of SO2 and growth of particles corrected for chamber wall loss in the control experiments? Were the wall loss controls done at different RH? How were the corrections performed? What are the uncertainties associated with correcting or not correcting for wall effects?

Methods – how was the SO2 analyzer calibrated? Did the authors have a NIST-traceable SO2 standard? What is the uncertainty in SO2 concentration that propagates into ySO2?

205 In the iodometric test using H2O2 as a standard, it is known that the reaction between H2O2 and KI might be complete after one hour but the reaction of organic peroxides and KI may take several hours up to a day (depending on the structure of the organic peroxide). As the authors have organic peroxide standards – I am curious why the authors decide to use H2O2 instead of organic peroxides? For future works, I suggest the authors to see for themselves how long the reaction takes to come to completion for their organic peroxides by following it after several hours. Another problem is the notorious difficulty of reproducing results – were replicates performed?

236 Can the authors discuss aerosol liquid water trends vs RH for the types of malonic/AS aerosols they are studying? There are also several hygroscopicity studies for SOA pure and mixed.

242 This can also be due to the ionic strength effects the authors talked about earlier

Fig S7 and General – Vapor pressure considers the partitioning between the gaseous form of a compound and its pure solid/liquid form. As the authors are considering partitioning between gas and water (e.g., Fig S7 plots data/estimations at RH 50%), wouldn't Henry's Law be a more appropriate parameter?

Fig. S7 and throughout the text – I see that 2-butanone peroxide actually has three peroxide (-OO-) moieties from what is shown in figure 1 and from its Sigma Aldrich

page? Two of those moieties are hydroperoxide (-OOH), and one is an ROOR. So why is the "-OO-" content for 2-butanone peroxide 2 instead of 3. If the authors only consider the hydroperoxide groups (-OOH) then the figure and text (and discussion) should be amended to clarify this and discuss why the interior -OO- isn't important.

304 The authors have an estimation of peroxide content in the particles and an measure of peroxide content in atomizer solution, so can the authors estimate how much of each peroxide stays in the condensed phase instead of assuming it all does? The assumption that all peroxides are nonvolatile seems to be in violation of the authors' statement in line 282 "measured $\gamma$SO2 depends both on reactivity and gas-particle partitioning of the organic peroxides."

325 Can you give some more information about why the H+ would be in the organic phase and not in the aqueous/inorganic phase?

348 Something is a bit confusing with Reaction 5 I think. If b is a quantity <1, then perhaps the ROH should also have some (1-b) multiplier? And suggest to limit the equation to ROOH for accounting purposes, and/or denote "ROOR" as ROOR' and show where R' goes too.

364 Is this only a factor of ionic strength? There seems to be some indication that droplets have a gradient in pH with the most acidic part at the interface, even for larger buffered aqueous droplets. Perhaps this discussion can be expanded to include this citation. https://www.pnas.org/content/pnas/115/28/7272.full.pdf

Section 3.4. Can the authors add in a discussion of what protons at the air-liquid interface can do to oxidize SO2 in addition to the Reactions 5-7? For example Hung and Hoffman shows a number of other dark reactions on acidic microdroplet surfaces including proposed radical formation. https://pubs.acs.org/doi/abs/10.1021/acs.est.5b01658

396 is it necessary to refer to pH in two different ways "increasing proton concentrations

(decreasing pH)"? Ions in hydrated aerosol mixtures should be talked about in terms of activity anyway, instead of concentration.

396, 467 and Figure 5. To be honest there does not seem to be much of a trend of ySO4 with pH that can support the statements (line 396)"The reactive uptake coefficient was found to increase with increasing proton concentrations (decreasing pH), which is consistent with acid-catalyzed reactions between peroxides and dissolved SO2 as measured in the bulk phase (Lind et al., 1987; Wang et al., 2019)" and in Line 467 "Increasing the condensed-phase acidity enhances the heterogeneous rate constant at low pH range." The authors agree that there is a "weak dependence on pH" (467) but the statements quoted here read quite strong, so the text then reads somewhat contradictory. From Table S1, I see that the pH experiments are 17, 23-35 has corresponding ySO2 range of 3.1 – 4.6 e3. Are you sure the margins of error in the ySO2 measurements and E-AIM modeling (both Y and X direction) are not larger than +/- roughly 20% from the mean? I believe calibration uncertainty in SO2 alone can get you there, not to mention acid estimations that are notoriously difficult and can be off by orders of magnitude. I don't doubt that in reality there may be a weak dependence, but I mainly want to see statements backed up by the data. Please (1) add uncertainty bars to figures, (2) temper the statements to say "may enhance" or "was found to weakly increase", and (3) acknowledge that within uncertainties, there may not be an observable trend here. I applaud the authors for acknowledging that they cannot fully explain pH trends, as there is a lot going in aerosol particles and we don't know what we don't know.

---

## Referee Comment (RC2) · Anonymous Referee #2 · 20 Jan 2021

Wang et al. ran a series of laboratory experiments to explore the uptake of SO2 onto aerosols containing organic peroxides. They systematically explored several factors, including RH, peroxide types, peroxide content, and aerosol pH. This study addresses an important topic, and the experiments provide insights into the factors that control the heterogenous conversion of SO2 to sulfate. This study is well within scope of the journal. My comments are below.

Major comments:

1. How good was the reproducibility of the experiments (data shown in Figure 2-6)? I am a little concerned about the small statistics in these experiments that the authors used to conclude any trend. Were there any replicate experiments done?

[Figure]

Minor comments:

1. Were the experiments conducted in a dark chamber? Could peroxides undergo photolysis?

2. Line 187: Does the repartitioning of SO2 from the wall depend on the type of organic peroxide in the chamber?

3. Line 208-line 209: "The average molecular mass for aerosol was assumed based on the chemical composition in order to calculate the molar fraction of total peroxides". The authors need to provide more details on how this was done, especially for the SOA particles. How were the chemical composition determined for SOA? What were the molar fractions of peroxides in the SOA particles?

4. Figure S9: the residual distribution does not look like a normal distribution.

5. When using the SMPS to derive the average aerosol surface area, how well was the RH maintained in the SMPS flow? In other words, could there be a size change due to a change in RH in the SMPS that leads to an underestimation of the surface area?

6. Could SO2 interacts with peroxides on the wall during the experiments? This includes the peroxides in the particles deposited on the wall and the gas-phase peroxides that were deposited on the wall.

---

## Author Comment (AC1) · 9 Feb 2021

**Response to anonymous referee #1**

We appreciate the constructive and informative comments from the reviewer.

Our response and corresponding revisions are listed below.

**General comments**

The authors studied the uptake coefficients of sulfur dioxide on particles containing three model organic peroxides (tert-butyl hydroperoxide, cumene hydroperoxide, and 2-butanone peroxide) as a function of (a) relative humidity (as proxy for particle liquid water), (b) particle acidity, and (4) composition of the particles (e.g., with malonic acid, or ammonium sulfate, or various model SOA material generated under dry conditions). The $SO_2$ was measured by a commercial analyzer and the particles were measured by SMPS. The pH was modeled by E-AIM. The methods are sound, and the paper is well written, and the discussion is fairly thorough. Moreover, the results are likely important for global modeling to better understand the atmospheric sulfur cycle. I request minor revisions based on the specific comments below.

**Specific comments** (line number precedes comment)

**Line 12** The terminology is a bit confusing. As I understand it "multifunctional" means multiple different functional groups (e.g., an alcohol and a hydroperoxide on the same compound) not multiple peroxide groups. Perhaps multiple peroxide groups on a compound would be better described as poly-peroxide (similarly to polyol) or just multiple peroxide. Please clarify this throughout the text. Also please be specific throughout the text whether you are referring to hydroperoxide moieties or all peroxides.

**Response**:

Thank you for the suggestions. We have modified the manuscript.

Line 10-11

"…to $10^{-2}$ at RH 71% as measured for an organic peroxide with multiple O-O groups…"

Line 12

"…organic peroxides with multiple peroxide groups have a higher $\gamma_{SO2}$…"

Line 286-288

"…higher $\gamma_{SO2}$ can be expected for organic peroxides with multiple O-O groups, lower vapour pressures and higher aqueous phase reactivities…"

The peroxide content measured by the KI method is for all types of peroxides ($H_2O_2$, ROOH, and ROOR) (Dotcherty et al., 2005). We have also clarified the related description in the manuscript.

Line 203-204

"…The total particulate peroxide content ($H_2O_2$, ROOH and ROOR) in these samples…"

**Line 12** As the authors only studied three peroxides, and they are not analogues in the way that would make the hydroperoxide moiety dependence clear, I would suggest against generalizing with this statement. At least the authors should add "in this study" to the statement to avoid overly broad generalizations or revise in another way.

**Response:**

Thanks for pointing this out. We have made the clarifications.

Line 11-12

"…Under similar conditions, the kinetics in this study were found to be structurally dependent…"

**Line 39** Mauldin et al 2012 did not positively identify stabilized Criegees, they suggested that it was a "Compound X" or "Unexplored oxidant X" that they believe to be SCIs. However, kinetic competition studies between SCI and water vapor vs $SO_2$ found that SCI + $SO_2$ is not competitive in the atmosphere for the dominant SCI $CH_2OO$ (Newland et al, ACP 2015, Nguyen et al, PCCP 2016). It is not clear which rates are used in Liu but that study seems to back up the previous lab work, as Nguyen et al estimated that $CH_2OO$ alone would be responsible for <6% $SO_2$ oxidation at a Southeast US site. I suggest to change this sentence to "sCIs were hypothesized to oxidize..." and please acknowledge the works before Liu 2019 that have already shown this pathway to be non-competitive at realistic RH using lab studies.
https://core.ac.uk/download/pdf/267289280.pdf
https://pubs.rsc.org/en/content/articlepdf/2016/cp/c6cp00053c

**Response:**

We thank the reviewer's comments. The corresponding part in the manuscript has been modified. The two related references have been added.

Line 39-43

 "…Stabilized Criegee intermediates (sCIs) were hypothesized to oxidize $SO_2$ rapidly and potentially serve as an important source of ambient sulfate (Mauldin et al., 2012). In the work by Newland et al. (2015) and Nguyen et al. (2016), this sCIs pathway was shown to play a minor role in sulfate formation. More recently, when Liu et al. (2019) applied this mechanism and kinetics to…"

**Line 115** (and elsewhere) The authors should insert the SI table number explicitly after the Experiment numbers so the reader can know where to look up the experiments.

**Response:**

The SI table number has been added after all the experiment numbers.

**Line 127** Please state the "different amounts" of HCl added for each experiment and the estimated particle pH that the different amounts of HCl correspond to. The authors say later that they estimate particle pH using E-AIM but this is worth mentioning in methods briefly first.

**Response:**

Thank you. Different amount of HCl, modeled pH and the E-AIM method have been added.

Line 130-134

"...different amounts of HCl (37%, Sigma-Aldrich) were added into the solution (0, 0.00002 M, 0.0001 M, 0.001 M HCl) prior to atomization. The initial pH of aerosol (2.5, 2.2, 1.6, 1, respectively) were modeled using E-AIM III model (Clegg et al., 1998) based on the initial molar ratios of inorganic species ($H^+$, $NH_4^+$, $SO_4^{2-}$, $Cl^-$) in the atomizing solution and measured RH (around 50%)..."

**Line 184** Were the losses of $SO_2$ and growth of particles corrected for chamber wall loss in the control experiments? Were the wall loss controls done at different RH? How were the corrections performed? What are the uncertainties associated with correcting or not correcting for wall effects? Methods – how was the $SO_2$ analyzer calibrated? Did the authors have a NIST traceable $SO_2$ standard? What is the uncertainty in $SO_2$ concentration that propagates into $y_{SO2}$?

**Response:**

We thank the reviewer for the comments.

Particle wall loss was corrected by assuming pseudo first-order loss rate in all $y_{SO2}$ measurements (Table S1). For all the other control experiments (Fig. S3-S6), $y_{SO2}$ was not calculated such that particle wall loss correction was not performed. Based on $SO_2$ wall loss control experiments (Fig. S6) under both dry (RH 28%) and humid (RH 74%) conditions, we did not observe any $SO_2$ wall loss but only $SO_2$ repartitioning from the wall due to the method we used here for measuring $y_{SO2}$. The repartitioning rate of $SO_2$ was thus corrected for $y_{SO2}$ measurements under high RH conditions (Exp. 14, RH>70%). The bias with/without correcting $SO_2$ repartitioning was found to be 40%.

The $SO_2$ analyzer (Model 43i, Thermo Fisher Scientific) was calibrated using a Multi-Gas Calibrator (Model 146i, Thermo Fisher Scientific) and a standard gas mixture (32 ppm $SO_2$, 610 ppm CO and 10.06% $CO_2$ balanced in $N_2$, Linde). The accuracy was estimated to be 1% full scale. $y_{SO2}$ was solved from equation (1) $-\frac{d[SO_2]}{dt} = \frac{1}{4}\gamma_{SO_2}A\bar{c}[SO_2]$, where the uncertainties in measured $y_{SO2}$ can be propagated from both $[SO_2]$ and A. The uncertainties have been estimated for each experiment and updated in Table S1.

**Line 205** In the iodometric test using $H_2O_2$ as a standard, it is known that the reaction between $H_2O_2$ and KI might be complete after one hour but the reaction of organic peroxides and KI may take several hours up to a day (depending on the structure of the organic peroxide). As the authors have organic peroxide standards – I am curious why the authors decide to use $H_2O_2$ instead of organic peroxides?

For future works, I suggest the authors to see for themselves how long the reaction takes to come to completion for their organic peroxides by following it after several hours. Another problem is the notorious difficulty of reproducing results – were replicates performed?

**Response:**

We agree that some studies use $H_2O_2$ as calibration standard for total peroxide quantification (Nguyen et al., 2010;Mutzel et al., 2013;Epstein et al., 2014;Krapf et al., 2016). Meanwhile, other studies also used organic peroxides as calibration standards in quantifying total peroxides in SOA (Docherty et al., 2005;Surratt et al., 2006).

In the current study, we used tert-butyl peroxide instead of $H_2O_2$ as a calibration standard since organic peroxides are the major peroxide type presented in SOA. In our KI method, KI was added in excess (> 100 times) so that we make sure the reaction proceeds at a relatively fast rate and completes within the 1-hour incubation. The measurement of peroxide content was performed in triplicates as indicated in the following figure. We thank the reviewer for the kind suggestion, future studies should be designed to investigate the different KI reactivities (oxidative potential) of organic peroxides with different molecular structures.

[Figure]

**Line 236** Can the authors discuss aerosol liquid water trends vs RH for the types of malonic/AS aerosols they are studying? There are also several hygroscopicity studies for SOA pure and mixed.

**Response:**

Thank you very much for the suggestion.

The aerosol liquid water content was found to be higher for ammonium sulfate aerosol than malonic acid aerosol. The hygroscopicity of ammonium sulfate is also higher than that of malonic acid. Adding inorganic component with higher hygroscopicity was proved to enhance the hygroscopicity of SOA based on the ZSR model (Varutbangkul et al., 2006;Meyer et al., 2009). As a result, the aerosol water content as well as hygroscopicity of organic peroxide containing ammonium sulfate aerosol can be estimated to be higher than organic peroxide containing malonic acid aerosol under similar RH conditions in the present study. However, the dissociation of ammonium sulfate is more extensive than malonic acid, which results in a significantly higher ionic strength for ammonium sulfate (40 mol kg$^{-1}$) than that of malonic acid (0.45 mol kg$^{-1}$) as indicated by EAIM III model under RH 50%.

[Figure]

Line 352-354

"…Although the aerosol water content for ammonium sulfate aerosol was found to be higher than that of malonic acid aerosol…"

**Line 242** This can also be due to the ionic strength effects the authors talked about earlier Fig S7 and General – Vapor pressure considers the partitioning between the gaseous form of a compound and its pure solid/liquid form. As the authors are considering partitioning between gas and water (e.g., Fig S7 plots data/estimations at RH 50%), wouldn't Henry's Law be a more appropriate parameter?

**Response:**

As suggested by the reviewer, we agree that Henry's law constant could be a more appropriate parameter for predicting the partitioning behavior between the gas phase and the liquid solution. However, to the best our knowledge, Henry's law constants for these organic peroxides are not available in the literature. Additionally, effective Henry's law constant should be considered here since condensed phase reactions (such as dissociation) may also occur and vary the theoretical Henry's law constants measured for pure peroxides. Since vapor pressure estimation methods (e.g. SIMPOL) are available, we used vapour pressure to represent the partitioning capacity of the volatile organic peroxide. However, it is important for future studies to investigate the effective Henry's law constant of different types organic peroxides in aerosol liquid water during the $SO_2$ reactive uptake process.

**Fig. S7 and throughout the text** – I see that 2-butanone peroxide actually has three peroxide (-OO-) moieties from what is shown in figure 1 and from its Sigma Aldrich page?

Two of those moieties are hydroperoxide (-OOH), and one is an ROOR. So why is the "-OO-" content for 2-butanone peroxide 2 instead of 3. If the authors only consider the hydroperoxide groups (-OOH) then the figure and text (and discussion) should be amended to clarify this and discuss why the interior -OO- isn't important.

**Response:**

Thank you for the suggestion. Both types of peroxides (ROOR and ROOH) are reactive towards S(IV). Thus, we refer both ROOH and ROOR in the manuscript.

Fig.S7 has been modified.

[Figure]

**Line 304** The authors have an estimation of peroxide content in the particles and an measure of peroxide content in atomizer solution, so can the authors estimate how much of each peroxide stays in the condensed phase instead of assuming it all does? The assumption that all peroxides are nonvolatile seems to be in violation of the authors' statement in line 282 "measured $\gamma_{SO2}$ depends both on reactivity and gas-particle partitioning of the organic peroxides."

**Response:**

Thank you for the suggestion. In the present study, the KI measured peroxide content indicates the potential for gas-to-particle partitioning of different types of organic peroxides. The particulate peroxide content was measured offline after filter collection from the chamber, which could be an underestimation of aerosol peroxide content during the chamber experiments. Based on the reviewer's comment, we have modified Fig. S8, where the predicted $\gamma_{SO2}$ has an uncertainty range due to the uncertainty in particulate peroxide content. The uncertainty in peroxide content (shadowed area) was estimated as the difference between the initial peroxide content (assuming no partitioning) and the KI measured peroxide content for 2-butanone peroxide collected from the chamber. In the future, an online method for aerosol phase peroxide quantification should be developed and used.

[Figure]

**Line 325** Can you give some more information about why the $H^+$ would be in the organic phase and not in the aqueous/inorganic phase?

**Response:**

Previous studies have shown aerosol undergo liquid-liquid phase separation can still have limited amount of water presented in the organic-rich phase (Dallemagne et al., 2016;Renbaum-Wolff et al., 2016). $H^+$ dissolved in aerosol water can then exist in the organic phase so that the organic-rich phase could be acidic. However, more aerosol water as well as $H^+$ would distribute in the inorganic phase due to its higher hygroscopicity. The statement in the manuscript was not focused on the distribution of $H^+$ in the organic phase. Instead, it described that with liquid-liquid separation, less water would be distributed in the organic phase such that the peroxide concentration could be higher than what we estimated assuming both peroxide and the aerosol water were evenly distributed in the homogeneous particle. A higher peroxide concentration

could be one of the potential reasons for the underestimation of model predicted $\gamma_{SO2}$ comparing to the experimentally measured $\gamma_{SO2}$ if liquid-liquid phase separation occurs in the current study.

**Line 348** Something is a bit confusing with Reaction 5 I think. If b is a quantity <1, then perhaps the ROH should also have some (1-b) multiplier? And suggest to limit the equation to ROOH for accounting purposes, and/or denote "ROOR" as ROOR' and show where R' goes too.

**Response:**

Thank you. We have modified the reaction mechanisms as following:

$$HO-\overset{\overset{O}{\|}}{\underset{O^-}{S}} + ROOH \overset{K}{\rightleftharpoons} \beta \overset{O^-}{\underset{O}{S}}-O-O-R + (1-\beta)\ SO_4^{2-} + (1-\beta)\ ROH + \beta\ H_2O + (1-\beta)\ H^+ \quad (5)$$

$$O^- - \overset{\cdot\cdot}{\underset{\overset{\|}{O}}{S}} - O - O - R + H^+ \overset{K_a^{-1}}{\rightleftharpoons} HO - \overset{\overset{O}{\|}}{\underset{\overset{\|}{O}}{S}} - O - R \quad (6)$$

$$HO - \overset{\overset{O}{\|}}{\underset{\overset{\|}{O}}{S}} - O - R \overset{k}{\longrightarrow} O - \overset{\overset{O}{\|}}{\underset{\overset{\|}{O}}{S}} - O - R + H^+ \quad (7)$$

**Line 364** Is this only a factor of ionic strength? There seems to be some indication that droplets have a gradient in pH with the most acidic part at the interface, even for larger buffered aqueous droplets. Perhaps this discussion can be expanded to include this citation. https://www.pnas.org/content/pnas/115/28/7272.full.pdf

**Response:**

We thank the reviewer for the suggestion. Besides ionic strength, we think interfacial properties, is also likely to explain the discrepancy between the aqueous phase and aerosol phase kinetics we observed in the current study. Wei et al. (2018) was cited in the following discussion where we think high ionic strength in the aerosol phase could accelerate the interfacial chemistry by the partitioning of anions to the air-liquid interface and promote the overall reaction kinetics via proton transfer mechanism.

However, we did not show any direct evidence in the current study indicating the relationship between the interfacial properties and $\gamma_{SO2}$. The particle studied in Wei et al. (2018) is significantly larger (20 μm) than the submicron particles (~200 nm) studied in the current work.

Previous studies showed evidences that aerosol pH is associated with the particle size (Craig et al., 2018;Keene et al., 2004;Young et al., 2013;Fang et al., 2017;Ding et al., 2019;Pye et al., 2020). Particles with different sizes could have different chemical compositions, hygroscopicity and mass transfer equilibrium timescales for the presented species. Additionally, the aerosol droplet in Wei et al. (2018) is phosphate-buffered, which has an overall pH above 7 where they observed the stable proton gradient while the aerosol pH range studied in our work is below 4. With acidic aerosol, it is uncertain if the distribution of protons would still exhibit a gradient with the highest concentration at the interface. As a result, we do not find it appropriate to extrapolate the results from Wei et al. (2018) to our study. Moreover, we observed a weak dependence of $\gamma_{SO2}$ on pH. There is less than one order of magnitude increase in $\gamma_{SO2}$ when pH decreased by 2 units. Thus, we concluded that interfacial chemistry could be one of the potential explanations for the enhanced kinetics observed in aerosol phase compared to bulk solution chemistry, but more direct evidences would be needed in future studies.

We have modified the discussion in the manuscript:

Line 389-398

"…In a recent study by Wei et al. (2018), a pH gradient was observed for phosphate-buffered aerosol droplets with the proton accumulated at the interface. Base on the pH-dependent aqueous phase kinetics measured in our previous work (Wang et al., 2019), such interfacial proton accumulation could potentially explain the enhanced kinetics we observed for aerosol in the current study. However, the chemical compositions are quite different. While phosphate-buffered particles were studied in Wei et al. (2018), acidic ammonium sulfate aerosol was used in our study. Also, the particle size in Wei et al. (2018) is significantly larger (20 μm) than what was studied in the current study (200 nm). Thus, it should be noted that there is no direct evidence from the current study showing the relationship between the interfacial properties and $\gamma_{SO2}$, and future studies are warranted…"

**Section 3.4** Can the authors add in a discussion of what protons at the air-liquid interface can do to oxidize $SO_2$ in addition to the Reactions 5-7? For example, Hung and Hoffman shows a number of other dark reactions on acidic microdroplet surfaces including proposed radical formation. https://pubs.acs.org/doi/abs/10.1021/acs.est.5b01658

**Response:**

Thank you for the comments. The protons at the aerosol interface can be effective in sulfate formation via multiple mechanisms. Protons can catalyze the rearrangement of the transient intermediate, peroxymonosulfite, formed in the peroxide S(IV) oxidation pathway. Besides, Hung et al. (2015) observed sulfite radical that can form sulfate via radical propagation chain,

which has significant signal at the acidic microdroplets. The formation of the sulfite radical needs the pre-existence of other radical species, where decomposition of organic peroxides in our system could be the source of hydroxy radical (Tong et al., 2016). Currently, we are not sure which mechanism plays the dominant role, and future studies are warranted to investigate this. We have added the following discussion to the manuscript:

Line 381-389

"…In addition to the catalytic effects of protons indicated in Eqn.5-7, Hung et al. (2015; 2018) observed significant $SO_3^{\cdot-}$ signal at the acidic microdroplet surface, which can promote sulfate formation via radical propagation chain initiated by surrounding radicals and molecular oxygen (Eqn. 8-11).

$$HSO_3^- + {}^{\cdot}OH \longrightarrow SO_3^{\cdot-} + H_2O \tag{8}$$
$$SO_3^{\cdot-} + O_2 \longrightarrow SO_5^{\cdot-} \tag{9}$$
$$SO_5^{\cdot-} + SO_3^{2-} \longrightarrow SO_4^{2-} + SO_4^{\cdot-} \tag{10}$$
$$SO_4^{\cdot-} + SO_3^{2-} \longrightarrow SO_4^{2-} + SO_3^{\cdot-} \tag{11}$$

Where the hydroxy radical can potentially be produced from decomposition of the labile organic peroxide in our system (Tong et al., 2016). However, we cannot distinguish whether the interfacial protons promote sulfate formation by catalyze the peroxide S(IV) oxidation pathway or the sulfur radical pathway at the current stage…"

**Line 396** is it necessary to refer to pH in two different ways "increasing proton concentrations (decreasing pH)"? Ions in hydrated aerosol mixtures should be talked about in terms of activity anyway, instead of concentration.

**Response:**

We have simplified the corresponding sentence.

Line 425
"…The reactive uptake coefficient was found to weakly increase with decreasing pH…"

**Line 396, 467** and **Figure 5**. To be honest there does not seem to be much of a trend of $y_{SO2}$ with pH that can support the statements (line 396)"The reactive uptake coefficient was found to increase with increasing proton concentrations (decreasing pH), which is consistent with acid-catalyzed reactions between peroxides and dissolved

SO$_2$ as measured in the bulk phase (Lind et al., 1987; Wang et al., 2019)" and in
**Line 467** "Increasing the condensed-phase acidity enhances the heterogeneous rate
constant at low pH range." The authors agree that there is a "weak dependence on
pH" (467) but the statements quoted here read quite strong, so the text then reads
somewhat contradictory. From Table S1, I see that the pH experiments are 17, 23-35
has corresponding y$_{SO2}$ range of 3.1 – 4.6 e3. Are you sure the margins of error in the
y$_{SO2}$ measurements and E-AIM modeling (both Y and X direction) are not larger than
+/- roughly 20% from the mean? I believe calibration uncertainty in SO$_2$ alone can get
you there, not to mention acid estimations that are notoriously difficult and can be off
by orders of magnitude. I don't doubt that in reality there may be a weak dependence,
but I mainly want to see statements backed up by the data. Please (1) add uncertainty bars to
figures, (2) temper the statements to say "may enhance" or "was found to
weakly increase", and (3) acknowledge that within uncertainties, there may not be an
observable trend here. I applaud the authors for acknowledging that they cannot fully
explain pH trends, as there is a lot going in aerosol particles and we don't know what
we don't know.

**Response:**

Thank you very much for the comment.
We have added the uncertainties to each measurement (Figure 2-6, Table S1) and discussed the
potential uncertainties in pH predictions. The corresponding discussions have been modified in
the manuscript.

Line 425-427

"…The reactive uptake coefficient was found to weakly increase with decreasing pH, which is
consistent with acid-catalyzed reactions between peroxides and dissolved SO$_2$ as measured in the
bulk phase…"

Line 446-450

"…It should also be noted that there are substantial uncertainties in estimating pH values,
originating from the partitioning of organics, organic-inorganic phase separations, mixing state
of specific ions, uncertain activity coefficients and the propagation of RH uncertainties (Clegg et
al., 2008; Fountoukis et al., 2009; Guo et al., 2016)…"

Line 498-499

"…Increasing the condensed-phase acidity may enhance the heterogeneous rate constant at low
pH …"

"…Also, it is likely that within the uncertainties, there may not be an observable $\gamma_{SO2}$-pH trend. Currently, we are not able to fully explain the pH dependence, and further studies are warranted…"

**References**

Craig, R. L., Peterson, P. K., Nandy, L., Lei, Z., Hossain, M. A., Camarena, S., Dodson, R. A., Cook, R. D., Dutcher, C. S., and Ault, A. P.: Direct determination of aerosol pH: Size-resolved measurements of submicrometer and supermicrometer aqueous particles, Anal. Chem., 90, 11232–11239, https://doi.org/10.1021/acs.analchem.8b00586, 2018.

Dallemagne, M. A., Huang, X. Y., and Eddingsaas, N. C.: Variation in pH of model secondary organic aerosol during liquid–liquid phase separation, J. Phys. Chem. A, 120, 2868-2876, https://doi.org/10.1021/acs.jpca.6b00275, 2016.

Ding, J., Zhao, P., Su, J., Dong, Q., Du, X., and Zhang, Y.: Aerosol pH and its driving factors in Beijing, Atmos. Chem. Phys., 19, 7939–7954, https://doi.org/10.5194/acp-19-7939-2019, 2019.

Docherty, K. S., Wu, W., Lim, Y. B., and Ziemann, P. J.: Contributions of organic peroxides to secondary aerosol formed from reactions of monoterpenes with $O_3$, Environ. Sci. Technol., 39, 4049-4059, 2005.

Epstein, S. A., Blair, S. L., and Nizkorodov, S. A.: Direct photolysis of α-pinene ozonolysis secondary organic aerosol: effect on particle mass and peroxide content, Environ. Sci. Technol., 48, 11251-11258, 2014.

Fang, T., Guo, H., Zeng, L., Verma, V., Nenes, A., and Weber, R. J.: Highly acidic ambient particles, soluble metals, and oxidative potential: A link between sulfate and aerosol toxicity, Environ. Sci. Technol., 51, 2611-2620, 10.1021/acs.est.6b06151, 2017.

Hung, H. M. and Hoffmann, M. R.: Oxidation of gas-phase $SO_2$ on the surfaces of acidic microdroplets: Implications for sulfate and sulfate radical anion formation in the atmospheric liquid phase, Environ. Sci. Technol., 49, 13768–13776, https://doi.org/10.1021/acs.est.5b01658, 2015.

Keene, W. C., Pszenny, A. A. P., Maben, J. R., Stevenson, E., and Wall, A.: Closure evaluation of size-resolved aerosol pH in the New England coastal atmosphere during summer, J. Geophys. Res.Atmos., 109(D23), D23202, doi:10.1029/2004JD004801, 2004.

Krapf, M., El Haddad, I., Bruns, E. A., Molteni, U., Daellenbach, K. R., Prévôt, A. S., Baltensperger, U., and Dommen, J.: Labile peroxides in secondary organic aerosol, Chem, 1, 603-616, 2016.

Meyer, N. K., Duplissy, J., Gysel, M., Metzger, A., Dommen, J.,Weingartner, E., Alfarra, M. R., Prev´ot, A. S. H., Fletcher, C., Good, N., McFiggans, G., Jonsson, A. M., Hallquist, M.,

Baltensperger, U., and Ristovski, Z. D.: Analysis of the hygroscopic and volatile properties of ammonium sulphate seeded and unseeded SOA particles, Atmos. Chem. Phys., 9, 721–732, http://www.atmos-chem-phys.net/9/721/2009/, 2009.

Mutzel, A., Rodigast, M., Iinuma, Y., Boege, O., and Herrmann, H.: An improved method for the quantification of SOA bound peroxides, Atmos. Environ., 67, 365–369, doi:10.1016/j.atmosenv.2012.11.012, 2013.

Nguyen, T. B., Bateman, A. P., Bones, D. L., Nizkorodov, S. A., Laskin, J., and Laskin, A.: High-resolution mass spectrometry analysis of secondary organic aerosol generated by ozonolysis of isoprene, Atmos. Environ., 44, 1032–1042, 2010.

Pye, H. O. T., Nenes, A., Alexander, B., Ault, A. P., Barth, M. C., Clegg, S. L., Collett Jr, J. L., Fahey, K. M., Hennigan, C. J., Herrmann, H., Kanakidou, M., Kelly, J. T., Ku, I. T., McNeill, V. F., Riemer, N., Schaefer, T., Shi, G., Tilgner, A., Walker, J. T., Wang, T., Weber, R., Xing, J., Zaveri, R. A., and Zuend, A.: The acidity of atmospheric particles and clouds, Atmos. Chem. Phys., 20, 4809-4888, 10.5194/acp-20-4809-2020, 2020.

Renbaum-Wolff, L., Song, M., Marcolli, C., Zhang, Y., Liu, P. F., Grayson, J. W., Geiger, F. M., Martin, S. T., and Bertram, A. K.: Observations and implications of liquid-liquid phase separation at high relative humidities in secondary organic material produced by α-pinene ozonolysis without inorganic salts, Atmos. Chem. Phys., 16, 7969–7979, https://doi.org/ 10.5194/acp16-7969-2016, 2016.

Surratt, J. D., Murphy, S. M., Kroll, J. H., Ng, N. L., Hildebrandt, L., Sorooshian, A., Szmigielski, R., Vermeylen, R., Maenhaut, W., Claeys, M., Flagan, R. C., and Seinfeld, J. H.: Chemical composition of secondary organic aerosol formed from the photooxidation of isoprene, J. Phys. Chem. A, 110, 9665–9690, 2006.

Tong, H., Arangio, A. M., Lakey, P. S. J., Berkemeier, T., Liu, F., Kampf, C. J., Brune, W. H., Pöschl, U., and Shiraiwa, M.: Hydroxyl radicals from secondary organic aerosol decomposition in water, Atmos. Chem. Phys., 16, 1761–1771, doi:10.5194/acp-16-1761-2016, 2016.

Varutbangkul, V., Brechtel, F. J., Bahreini, R., Ng, N. L., Keywood, M. D., Kroll, J. H., Flagan, R. C., Seinfeld, J. H., Lee, A., and Goldstein, A. H.: Hygroscopicity of secondary organic aerosols formed by oxidation of cycloalkenes, monoterpenes, sesquiterpenes, and related compounds, Atmos. Chem. Phys., 6, 2367–2388, http://www.atmos-chem-phys.net/6/2367/2006/, 2006.

Wei, H., Vejerano, E. P., Leng, W., Huang, Q., Willner, M. R., Marr, L. C., and Vikesland, P. J.: Aerosol microdroplets exhibit a stable pH gradient, P. Natl. Acad. Sci. USA, 115, 7272–7277, https://doi.org/10.1073/pnas.1720488115, 2018.

Young, A. H., Keene, W. C., Pszenny, A. A. P., Sander, R., Thornton, J. A., Riedel, T. P., and Maben, J. R.: Phase partitioning of soluble trace gases with size-resolved aerosols in near-surface continental air over northern Colorado, USA, during winter, J. Geophys. Res. Atmos., 118, 9414–9427, doi:10.1002/Jgrd.50655, 2013.

---

## Author Comment (AC2) · 9 Feb 2021

**Response to comments from referee #2**

We thank the reviewer for the constructive comments, with which we have addressed point-by-point and modified the manuscript accordingly as below.

**General comments**

Wang et al. ran a series of laboratory experiments to explore the uptake of $SO_2$ onto aerosols containing organic peroxides. They systematically explored several factors, including RH, peroxide types, peroxide content, and aerosol pH. This study addresses an important topic, and the experiments provide insights into the factors that control the heterogenous conversion of $SO_2$ to sulfate. This study is well within scope of the journal. My comments are below.

**Major comments:**

1. How good was the reproducibility of the experiments (data shown in Figure 2-6)? I am a little concerned about the small statistics in these experiments that the authors used to conclude any trend. Were there any replicate experiments done?

**Response**:

Thank you for the comments. Exp.10-12 (Table S1) were performed under similar conditions, and a standard deviation of 26% was found among the three different measurements. The trends of $\gamma_{SO_2}$ reported in Figure 4-6 were based on a log scale. The measured deviation is less likely to change our conclusions.

As for the accuracy, $\gamma_{SO_2}$ was solved from equation (1) $-\frac{d[SO_2]}{dt} = \frac{1}{4}\gamma_{SO_2}A\bar{c}[SO_2]$, where the uncertainties in measured $y_{SO2}$ can be propagated from estimated instrument accuracy in both A (particle surface area) and $[SO_2]$. The $SO_2$ analyzer (Model 43i, Thermo Fisher Scientific) was calibrated using a Multi-Gas Calibrator (Model 146i, Thermo Fisher Scientific) and a standard gas mixture (32 ppm $SO_2$, 610 ppm CO and 10.06% $CO_2$ balanced in $N_2$, Linde) with the accuracy estimated to be 1% of full scale. We have propagated the uncertainties for each experiment with corresponding updates in table S1 and Figure 2-6.

**Minor comments:**

1. Were the experiments conducted in a dark chamber? Could peroxides undergo photolysis?

**Response:**

Experiments in the current study were performed in the 1 m$^3$ chamber located in our lab, which is covered with a piece of black fabric. Thus, our experiments were performed under dark conditions with negligible effects from photolysis on peroxides.

2. Line 187: Does the repartitioning of $SO_2$ from the wall depend on the type of organic peroxide in the chamber?

**Response:**

Thanks for the comments. Different types of organic peroxides have different vapour pressures and reactivities towards $SO_2$, which might influence the repartitioning rate of $SO_2$ from the chamber wall during our experiments. We believe different types of organic peroxides might impact the repartitioning of $SO_2$ from the chamber wall, but it also depends on the amount of peroxides available during the experiments. We have performed $SO_2$-organic peroxide vapour experiments under similar RH conditions (TB peroxide, RH 27%; 2B peroxide, RH 28%). As shown in the figure below, the $SO_2$ repartitioning rate for 2B peroxide and TB peroxide have no significant difference under similar RH conditions/initial $SO_2$ mixing ratios.

[Figure]

However, a significant enhancement of the $SO_2$ repartitioning rate was observed when the relative humidity increased from 28% to 74% for 2B peroxide (similar initial mixing ratios). As a result, the potential effects coming from peroxide types on the $SO_2$ repartitioning rate could be much less significant than that from relative humidity under the experimental conditions in the current study.

3. Line 208-line 209: "The average molecular mass for aerosol was assumed based on the chemical composition in order to calculate the molar fraction of total peroxides". The authors need to provide more details on how this was done, especially for the SOA particles. How were the chemical composition determined for SOA? What were the molar fractions of peroxides in the SOA particles?

**Response**:

Since the chemical composition of SOA is more complicated than the peroxide/ammonium sulfate aerosol, there exist large uncertainty in estimating the molar of total SOA molecules. As a result, the current study didn't measure the peroxide molar fraction in SOA samples. In this study, we measured the molar fraction of peroxides in the peroxide/ammonium sulfate mixed aerosol. Based on the initial mixing ratio (2:1) and the molecular mass of peroxide/ammonium sulfate, we can estimate the averaged molar of the aerosol as:

$$\text{Molar fraction of peroxide} = \frac{N_{peroxide}}{N_{aerosol}} = N_{peroxide} \frac{M_{(NH_4)_2SO_4} f_{(NH_4)_2SO_4} + M_{peroxide} f_{peroxide}}{m_{aerosol}}$$

where $m_{aerosol}$ is the weighed aerosol mass on the filter; $M_{(NH_4)_2SO_4}$ and $M_{peroxide}$ are the molecular mass of ammonium sulfate and peroxide, respectively; $f_{(NH_4)_2SO_4}$ and $f_{peroxide}$ are the initial molar fraction of ammonium sulfate and peroxide; $N_{peroxide}$ and $N_{aerosol}$ are the measured peroxide molar and calculated aerosol molar, respectively. The corresponding information has been added the manuscript.

Line 215-224:

"…An average molecular mass for seed particles (SOA + ammonium sulfate) was assumed based on the chemical composition in order to calculate the molar fraction of total peroxides using the following equation:

$$\text{Molar fraction of peroxide} = \frac{N_{peroxide}}{N_{aerosol}} = N_{peroxide} \frac{M_{(NH_4)_2SO_4} f_{(NH_4)_2SO_4} + M_{peroxide} f_{peroxide}}{m_{aerosol}}$$

where $m_{aerosol}$ is the weighed aerosol mass on the filter; $M_{(NH_4)_2SO_4}$ and $M_{peroxide}$ are the molecular mass of ammonium sulfate and peroxide, respectively; $f_{(NH_4)_2SO_4}$ and $f_{peroxide}$ are the initial molar fraction of ammonium sulfate and peroxide; $N_{peroxide}$ and $N_{aerosol}$ are the measured peroxide molar and calculated aerosol molar, respectively..."

4. Figure S9: the residual distribution does not look like a normal distribution.

**Response**:

Thank you for the comment. A quantile-quantile plot can be made based on the multi-linear regression (MLR) built in the current study. The sample data should fall on the diagnostic line for an ideal normal distribution. An evenly distributed residual points can be found around the diagnostic line of the plot, indicating the normality of the residuals calculated from the MLR.

[Figure]

5. When using the SMPS to derive the average aerosol surface area, how well was the RH maintained in the SMPS flow? In other words, could there be a size change due to a change in RH in the SMPS that leads to an underestimation of the surface area?

**Response**:

The custom-built SMPS in our lab uses the recirculated excess flow as the sheath flow during the measurements. We typically start SMPS at the very beginning of each experiment to measure the background aerosol concentration inside the chamber. After sampling from the chamber continuously for at least 30 minutes, we expect that the recirculation flow has a RH similar to what inside the chamber. As a result, we do not expect water evaporation inside the SMPS has a significant impact during our measurements.

6. Could $SO_2$ interacts with peroxides on the wall during the experiments? This includes the peroxides in the particles deposited on the wall and the gas-phase peroxides that were deposited on the wall.

**Response**:

Thank you for the comments. There was no SO₂ decay when peroxide vapours were introduced into the chamber without particles under both low and high RH conditions, as shown in Figure S6a and S6b, respectively. This result indicates that there is no interaction between $SO_2$ and any gas-phase peroxides that immediately deposit on the chamber wall.

[Figure]

Another possibility is for peroxide/ammonium sulfate particles to deposit on the chamber wall, and for $SO_2$ to interact with deposited peroxides. However, most of the particles remain suspended (80-90%) during the $\gamma_{SO_2}$ measurements (<10 minutes) as indicated by the following SMPS data for Exp.10.

[Figure]

Also, we did not observe any significant $SO_2$ loss at the beginning of each experiment before introducing aerosol into the chamber. Thus, $SO_2$ loss caused by deposited peroxides from the

previous experiments can also be excluded. The chamber was flushed overnight between each experiment with zero air to minimize carryover.

---

## Author Response (AR2)

**Response to reviewer and editor comments**

We appreciate the constructive and informative comments from the editor and the reviewers.

Our response and corresponding revisions are listed below.

**Editor Comments**

**Page 3** "Reactive nitrogen species (such as NO2) have also been put forward to account for the missing sulfate at relatively high aerosol pH (close to 7) (Wang et al., 2016; Cheng et al., 2016)."

Cheng et al. (2016) suggested NO2 as a major contributor for the missing sulfate a pH range of  $\sim$ 5 to  $\sim$ 6. I would suggest being more specific about the pH range in Cheng et al. (2016) because "close to 7" might be misleading.

**Response**:**

Thank you for pointing this out. The pH range have been clarified.

**Line 65-67**

"...Reactive nitrogen species (such as  $NO_2$ ) have also been proposed as a dominant sulfate formation pathway when aerosol pH was estimated to be 5-6 in Cheng et al. (2016) and close to 7 in Wang et al. (2016) under severe haze scenarios..."

**Page 3** "However, such high aerosol pH is not substantiated by thermodynamic models, which conclude that pH ranges between 4 and 5 even in polluted regions (Song et al., 2018;Guo et al., 2017)".

For the aerosol pH range, later studies (e.g., Shi et al. 2017; Ding et al. 2019) show that the modelled aerosol pH in Beijing can go beyond the range of 4-5, even >6. I would suggest updating this and related statement. Note that Cheng et al. (2016) highlighted the importance of both  $NH_3$  and dust in regulating aerosol pH. Guo et al. (2017), however, didn't consider the contribution of dusts. Moreover, higher RH and higher aerosol concentrations may lead to further increase of aerosol pH (Zheng et al. 2020).

**Response**:**

Thank you for the suggestions. The corresponding discussions have been modified.

**Line 67-73**

"...While such high aerosol pH is not substantiated by some thermodynamic modeling results, which concluded that pH ranges between 4 and 5 in polluted regions (Song et al., 2018;Guo et al., 2017), other studies that highlighted the roles of ammonia and dust found aerosol pH could

be higher than 6 (Shi et al., 2017; Ding et al., 2019). Furthermore, higher aerosol water content and PM mass concentration in polluted areas have been shown to enhance aerosol pH via a multiphase buffering process (Zheng et al., 2020)..."

**References**

Ding, J., Zhao, P., Su, J., Dong, Q., Du, X., and Zhang, Y.: Aerosol pH and its driving factors in Beijing, Atmos. Chem. Phys., 19, 7939-7954, 10.5194/acp-19-7939-2019, 2019.

Shi, G., Xu, J., Peng, X., Xiao, Z., Chen, K., Tian, Y., Guan, X., Feng, Y., Yu, H., Nenes, A., and Russell, A. G.: pH of aerosols in a polluted atmosphere: Source contributions to highly acidic aerosol, Environ. Sci. Technol., 51, 4289-4296, 10.1021/acs.est.6b05736, 2017.

Zheng, G., Su, H., Wang, S., Andreae, M. O., Pöschl, U., and Cheng, Y.: Multiphase buffer theory explains contrasts in atmospheric aerosol acidity, Science, 369, 1374-1377, 10.1126/science.aba3719, 2020.

**Response to anonymous referee #1**

**General comments**

The authors studied the uptake coefficients of sulfur dioxide on particles containing three model organic peroxides (tert-butyl hydroperoxide, cumene hydroperoxide, and 2-butanone peroxide) as a function of (a) relative humidity (as proxy for particle liquid water), (b) particle acidity, and (4) composition of the particles (e.g., with malonic acid, or ammonium sulfate, or various model SOA material generated under dry conditions). The SO2 was measured by a commercial analyzer and the particles were measured by SMPS. The pH was modeled by E-AIM. The methods are sound, and the paper is well written, and the discussion is fairly thorough. Moreover, the results are likely important for global modeling to better understand the atmospheric sulfur cycle. I request minor revisions based on the specific comments below.

**Specific comments (line number precedes comment)**

**Line 12** The terminology is a bit confusing. As I understand it "multifunctional" means multiple different functional groups (e.g., an alcohol and a hydroperoxide on the same compound) not multiple peroxide groups. Perhaps multiple peroxide groups on a compound would be better described as poly-peroxide (similarly to polyol) or just multiple peroxide. Please clarify this throughout the text. Also please be specific throughout the text whether you are referring to hydroperoxide moieties or all peroxides.

**Response**:**

Thank you for the suggestions. We have modified the manuscript.

Line 31-32

"...to 10-2 at RH 71% as measured for an organic peroxide with multiple O-O groups..."

Line 33

"...organic peroxides with multiple peroxide groups have a higher  $\gamma_{SO2}...$  "

Line 315-317

"...higher  $\gamma_{SO2}$  can be expected for organic peroxides with multiple O-O groups, lower vapour pressures and higher aqueous phase reactivities..."

The peroxide content measured by the KI method is for all types of peroxides ( $H_2O_2$ , ROOH, and ROOR) (Dotcherty et al., 2005). We have also clarified the related description in the manuscript.

Line 229-230

"... The total particulate peroxide content (H2O2, ROOH and ROOR) in these samples..."

Line 12 As the authors only studied three peroxides, and they are not analogues in the way that would make the hydroperoxide moiety dependence clear, I would suggest against generalizing with this statement. At least the authors should add "in this study" to the statement to avoid overly broad generalizations or revise in another way.

**Response:**

Thanks for pointing this out. We have made the clarifications.

Line 32-33

"...Under similar conditions, the kinetics in this study were found to be structurally dependent..."

Line 39 Mauldin et al 2012 did not positively identify stabilized Criegees, they suggested that it was a "Compound X" or "Unexplored oxidant X" that they believe to be SCIs. However, kinetic competition studies between SCI and water vapor vs SO2 found that SCI + SO2 is not competitive in the atmosphere for the dominant SCI CH2OO (Newland et al, ACP 2015, Nguyen et al, PCCP 2016). It is not clear which rates are used in Liu but that study seems to back up the previous lab work, as Nguyen et al estimated that CH2OO alone would be responsible for <6%

SO2 oxidation at a Southeast US site. I suggest to change this sentence to "sCIs were hypothesized to oxidize..." and please acknowledge the works before Liu 2019 that have already shown this pathway to be non-competitive at realistic RH using lab studies. https://core.ac.uk/download/pdf/267289280.pdf https://pubs.rsc.org/en/content/articlepdf/2016/cp/c6cp00053c

**Response:**

We thank the reviewer's comments. The corresponding part in the manuscript has been modified. The two related references have been added.

**Line 60-64**

"...Stabilized Criegee intermediates (sCIs) were hypothesized to oxidize SO2 rapidly and potentially serve as an important source of ambient sulfate (Mauldin et al., 2012). In the work by Newland et al. (2015) and Nguyen et al. (2016), this sCIs pathway was shown to play a minor role in sulfate formation. More recently, when Liu et al. (2019) applied this mechanism and kinetics to a source-oriented WRF-Chem model..."

Line 115 (and elsewhere) The authors should insert the SI table number explicitly after the Experiment numbers so the reader can know where to look up the experiments.

**Response:**

The SI table number has been added after all the experiment numbers.

**Line 127** Please state the "different amounts" of HCl added for each experiment and the estimated particle pH that the different amounts of HCl correspond to. The authors say later that they estimate particle pH using E-AIM but this is worth mentioning in methods briefly first.

**Response:**

Thank you. Different amount of HCl, modeled pH and the E-AIM method have been added.

**Line 156-160**

"...different amounts of HCl (37%, Sigma-Aldrich) were added into the solution (0, 0.00002 M, 0.0001 M, 0.001 M HCl) prior to atomization. The initial pH of aerosol (2.5, 2.2, 1.6, 1,

respectively) were modeled using E-AIM III model (Clegg et al., 1998) based on the initial molar ratios of inorganic species ( $H^+$ ,  $NH_4^+$ ,  $SO_4^{2-}$ ,  $Cl^-$ ) in the atomizing solution and measured RH (around 50%)..."

Line 184 Were the losses of SO2 and growth of particles corrected for chamber wall loss in the control experiments? Were the wall loss controls done at different RH? How were the corrections performed? What are the uncertainties associated with correcting or not correcting for wall effects? Methods – how was the SO2 analyzer calibrated? Did the authors have a NIST traceable SO2 standard? What is the uncertainty in SO2 concentration that propagates into  $y_{SO2}$ ?

**Response:**

We thank the reviewer for the comments.

Particle wall loss was corrected by assuming pseudo first-order loss rate in all  $y_{SO2}$  measurements (Table S1). For all the other control experiments (Fig. S3-S6),  $y_{SO2}$  was not calculated such that particle wall loss correction was not performed. Based on SO2 wall loss control experiments (Fig. S6) under both dry (RH 28%) and humid (RH 74%) conditions, we did not observe any SO2 wall loss but only SO2 repartitioning from the wall due to the method we used here for measuring  $y_{SO2}$ . The repartitioning rate of SO2 was thus corrected for  $y_{SO2}$  measurements under high RH conditions (Expt. 14, RH>70%). The bias with/without correcting SO2 repartitioning was found to be 40%.

The SO2 analyzer (Model 43i, Thermo Fisher Scientific) was calibrated using a Multi-Gas Calibrator (Model 146i, Thermo Fisher Scientific) and a standard gas mixture (32 ppm SO2, 610 ppm CO and 10.06% CO2 balanced in N2, Linde). The accuracy was estimated to be 1% full scale.  $y_{SO2}$  was solved from equation (1)  $-\frac{d[SO_2]}{dt} = \frac{1}{4}\gamma_{SO_2}A\bar{c}[SO_2]$ , where the uncertainties in measured  $y_{SO2}$  can be propagated from both [SO2] and A. The uncertainties have been estimated for each experiment and updated in Table S1.

Line 205 In the iodometric test using  $H_2O_2$  as a standard, it is known that the reaction between  $H_2O_2$  and KI might be complete after one hour but the reaction of organic peroxides and KI may take several hours up to a day (depending on the structure of the organic peroxide). As the authors have organic peroxide standards – I am curious why the authors decide to use  $H_2O_2$  instead of organic peroxides?

For future works, I suggest the authors to see for themselves how long the reaction takes to come to completion for their organic peroxides by following it after several hours. Another problem is the notorious difficulty of reproducing results – were replicates performed?

**Response:**

We agree that some studies use  $H_2O_2$  as calibration standard for total peroxide quantification (Nguyen et al., 2010;Mutzel et al., 2013;Epstein et al., 2014;Krapf et al., 2016). Meanwhile, other studies also used organic peroxides as calibration standards in quantifying total peroxides in SOA (Docherty et al., 2005;Surratt et al., 2006).

In the current study, we used tert-butyl peroxide instead of  $H_2O_2$  as a calibration standard since organic peroxides are the major peroxide type presented in SOA. In our KI method, KI was added in excess (> 100 times) so that we make sure the reaction proceeds at a relatively fast rate and completes within the 1-hour incubation. The measurement of peroxide content was performed in triplicates as indicated in the following figure. We thank the reviewer for the kind suggestion, future studies should be designed to investigate the different KI reactivities (oxidative potential) of organic peroxides with different molecular structures.

Line 236 Can the authors discuss aerosol liquid water trends vs RH for the types of malonic/AS aerosols they are studying? There are also several hygroscopicity studies for SOA pure and mixed.

**Response:**

Thank you very much for the suggestion.

The aerosol liquid water content was found to be higher for ammonium sulfate aerosol than malonic acid aerosol. The hygroscopicity of ammonium sulfate is also higher than that of malonic acid. Adding inorganic component with higher hygroscopicity was proved to enhance the hygroscopicity of SOA based on the ZSR model (Varutbangkul et al., 2006;Meyer et al., 2009). As a result, the aerosol water content as well as hygroscopicity of organic peroxide containing ammonium sulfate aerosol can be estimated to be higher than organic peroxide containing malonic

acid aerosol under similar RH conditions in the present study. However, the dissociation of ammonium sulfate is more extensive than malonic acid, which results in a significantly higher ionic strength for ammonium sulfate (40 mol kg-1) than that of malonic acid (0.45 mol kg-1) as indicated by EAIM III model under RH 50%.

**Line 381-383**

"...Although the aerosol water content for ammonium sulfate aerosol was found to be higher than that of malonic acid aerosol..."

Line 242 This can also be due to the ionic strength effects the authors talked about earlier Fig S7 and General – Vapor pressure considers the partitioning between the gaseous form of a compound and its pure solid/liquid form. As the authors are considering partitioning between gas and water (e.g., Fig S7 plots data/estimations at RH 50%), wouldn't Henry's Law be a more appropriate parameter?

**Response:**

As suggested by the reviewer, we agree that Henry's law constant could be a more appropriate parameter for predicting the partitioning behavior between the gas phase and the liquid solution. However, to the best our knowledge, Henry's law constants for these organic peroxides are not available in the literature. Additionally, effective Henry's law constant should be considered here since condensed phase reactions (such as dissociation) may also occur and vary the theoretical Henry's law constants measured for pure peroxides. Since vapor pressure estimation methods (e.g. SIMPOL) are available, we used vapour pressure to represent the partitioning capacity of the volatile organic peroxide. However, it is important for future studies to investigate the

effective Henry's law constant of different types organic peroxides in aerosol liquid water during the SO2 reactive uptake process.

**Fig. S7 and throughout the text** – I see that 2-butanone peroxide actually has three peroxide (-OO-) moieties from what is shown in figure 1 and from its Sigma Aldrich page?

Two of those moieties are hydroperoxide (-OOH), and one is an ROOR. So why is the "-OO-" content for 2-butanone peroxide 2 instead of 3. If the authors only consider the hydroperoxide groups (-OOH) then the figure and text (and discussion) should be amended to clarify this and discuss why the interior -OO- isn't important.

**Response:**

Thank you for the suggestion. Both types of peroxides (ROOR and ROOH) are reactive towards S(IV). Thus, we refer both ROOH and ROOR in the manuscript.

Fig.S7 has been modified.

Line 304 The authors have an estimation of peroxide content in the particles and an measure of peroxide content in atomizer solution, so can the authors estimate how much of each peroxide stays in the condensed phase instead of assuming it all does? The assumption that all peroxides are nonvolatile seems to be in violation of the authors' statement in line 282 "measured  $\gamma_{SO2}$  depends both on reactivity and gas-particle partitioning of the organic peroxides."

**Response:**

Thank you for the suggestion. In the present study, the KI measured peroxide content indicates the potential for gas-to-particle partitioning of different types of organic peroxides. The particulate peroxide content was measured offline after filter collection from the chamber, which could be an underestimation of aerosol peroxide content during the chamber experiments. Based on the reviewer's comment, we have modified Fig. S8, where the predicted  $\gamma_{SO2}$  has an uncertainty range due to the uncertainty in particulate peroxide content. The uncertainty in peroxide content (shadowed area) was estimated as the difference between the initial peroxide content (assuming no partitioning) and the KI measured peroxide content for 2-butanone peroxide collected from the chamber. In the future, an online method for aerosol phase peroxide quantification should be developed and used.